

# The role of liquid water percolation representation to estimate snow water equivalent in a Mediterranean mountain region (Mount Lebanon)

Abbas Fayad[1,2] and Simon Gascoin[1]

[1]Centre d'Etudes Spatiales de la Biosphère (CESBIO), UPS/CNRS/IRD/CNES, Toulouse, France
[2]Now at the Centre for Hydrology, University of Saskatchewan, Saskatoon, Saskatchewan, Canada

*Correspondence to*: Abbas Fayad (abbas.fayad@usask.ca)

**Abstract.** In many Mediterranean mountain regions, the seasonal snowpack is an essential yet poorly known water resource. Here, we examine, for the first time, the spatial distribution and evolution of the snow water equivalent (SWE) during three snow seasons (2013-2016) in the coastal mountains of Lebanon. We run SnowModel (Liston and Elder, 2006a), a spatially-distributed, process-based snow model, at 100 m resolution forced by new automatic weather station (AWS) data in three snow-dominated basins of Mount Lebanon. We evaluate a recent upgrade of the liquid water percolation scheme in SnowModel, which was introduced to improve the simulation of the snow water equivalent (SWE) and runoff in warm maritime regions. The model is evaluated against continuous snow depth and snow albedo observations at the AWS, manual SWE measurements, and MODIS snow cover area between 1200 m and 3000 m a.s.l.. The results show that the new percolation scheme yields better performance especially in terms of SWE but also in snow depth and snow cover area. Over the simulation period between 2013 and 2016, the maximum snow mass was reached between December and March. Peak mean SWE (above 1200 m a.s.l.) changed significantly from year to year in the three study catchments with values ranging between 73 mm and 286 mm we (RMSE between 160 and 260 mm w.e.). We suggest that the major sources of uncertainty in simulating the SWE, in this warm Mediterranean climate, can be attributed to forcing error but also to our limited understanding of the separation between rain and snow at lower-elevations, the transient snow melt events during the accumulation season, and the high-variability of snow depth patterns at the sub-pixel scale due to the wind-driven blown-snow redistribution into karstic features and sinkholes. Yet, the use of a process-based snow model with minimal requirements for parameter estimation provides a basis to simulate snow mass SWE in non-monitored catchments and characterize the contribution of snowmelt to the karstic groundwater recharge in Lebanon. While this research focused on three basins in the Mount Lebanon, it serves as a case study to highlight the importance of wet snow processes to estimate SWE in Mediterranean mountain regions.

## 1 Introduction

Winter snowpack has a large influence over the hydrological processes in Mediterranean mountains and the spring meltwater from the seasonal snowpack often represent key water resource for downstream areas (Fayad et al., 2017a). In such regions,





large fraction of the annual precipitation falls during the winter season, which extends over a period of five months (Fayad et al., 2017a). With a limited precipitation season and warm and dry summers, the Mediterranean mountainous areas are among the most vulnerable to a warming climate (e.g, Nohara et al., 2006; Loarie et al., 2009; Kumar et al., 2016).

Our understanding of the snowpack hydrological contribution in many Mediterranean mountains remains incomplete. Snow monitoring in regions such as the central Chilean Andes, North Africa, and the Eastern Mediterranean Sea is relatively new
(Fayad et al., 2017a). The increased interest, over the past decade, in understanding snow accumulation and melt in the less monitored Mediterranean mountains is mainly driven by the need to meet the increased demand for water resources. Mediterranean mountain regions, which share many characteristics of the snow in maritime regions (Fayad et al., 2017a), are characterized by their deep multilayer winter snowpack, which can exceed the 3 meters in depth, coarser grain size due to wetting, almost isothermal winter snowpack, and the high sensitivity of the precipitation phase to air temperature (Sturm et
al., 1995; Sproles, et al., 2013; Pflug et al., 2019). Capturing the meltwater runoff dynamics in Mediterranean can be challenging especially when transient snowmelt events are common during the winter season.

To reduce the uncertainty in simulating the snow water equivalent (SWE) in warm maritime regions, Pflug et al. (2019) implemented a discrete representation of snow liquid water percolation into the snowpack using gravity drainage to replace the standard snow density threshold routine in the distributed snowpack model SnowModel (Liston and Elder, 2006a). The
new scheme by Pflug et al. (2019) showed improvement in simulating SWE compared to the default model (Liston and Elder, 2006a) which tended to strongly overestimate the SWE at a Snow Telemetry station (SNOTEL) in the Washington Olympic Mountains, USA. The default liquid water percolation scheme did not allow the release of meltwater runoff during the accumulation season, causing a large over-accumulation of SWE. This process is typical of warm Mediterranean snowpack, such as the Mount Lebanon snowpack.
In this study, we make use of a newly available snow dataset (Fayad et al., 2017b) to evaluate this new percolation scheme in SnowModel to compute the SWE distribution and its evolution, for the first time, in three basins in the warm Mediterranean mountains of Lebanon. We evaluate the model outputs using both in situ (continuous snow depth records at AWS and snow course measurements) and remote sensing data (MODIS snow cover area). We show that the improved percolation scheme led to better performance therefore, we used this model version to estimate the daily evolution of the SWE distribution over
three water years (2013-2014, 2014-2015 and 2015-2016, hereafter W1314, W1415, and W1516, respectively).

**2 Study area**

Lebanon has relatively abundant water resources (857 m3 per capita in 2014 according to World Bank data). Nevertheless, water shortages are frequent during the summer. The water scarcity became particularly acute in Lebanon between 1998 and 2012, when the worst drought of the past nine centuries hit the Eastern Mediterranean (Cook et al., 2016). The difficulty to
meet water demand in Lebanon is also exacerbated by the poor management of the water resources (Lebanese Ministry of



Energy and Water, 2012), the increase in urbanization, and population growth over the past decades (Lebanese Ministry of Environment, 2011).

In Lebanon, the water demand for agriculture is concentrated in the irrigated Bekaa Valley between the Mount Lebanon and the Anti Lebanon mountain ranges. On the other hand, the water demand in the coastal strip of Lebanon, between the Mediterranean Sea and Mount Lebanon, is mostly associated to domestic and industrial uses. This narrow coastal plain is heavily urbanized and includes the Beirut metropolitan area where approximately a third of the Lebanese population lives. In this area, the urban water supply relies on the discharge of springs and water wells from the highly karstified Jurassic limestone aquifers (Margane et al., 2013). The recharge area of this karstic system is located on Mount Lebanon, a mountain range which extends over a distance of 150 km parallel to the Mediterranean coast with a main plateau averaging 2500 m a.s.l. elevation and peak at 3088 m a.s.l.

Most of the precipitation occurs between January and March due to the influence of the Mediterranean climate and therefore areas above 1200 m a.s.l. receive between 50 to 67% of their total annual precipitation as snow (Shaban et al., 2004; Aouad-Rizk et al., 2005; Mhawej et al., 2014; Telesca et al., 2014). Winter precipitation events are due to the orographic lifting of the cold northerly currents from the Anatolian plateau (Fish, 1944). Snowmelt is a major contributor to the groundwater recharge (Margane et al., 2013; UNDP, 2014). In this upper cretaceous formation, usually above 1600 m a.s.l., snowmelt contributes to roughly 75% of the groundwater recharge (Figure 1) (Margane et al., 2013; Königer and Margane, 2014).

This study focuses on the upper catchment areas of the Abou Ali, Ibrahim, and El Kelb river basins (Figure 1; Table 1). These basins provide key water resources to two major Governorates – the northern districts of Mount Lebanon Governorate and the districts of the North Governorate – as well as the densely populated coastal cities of Lebanon, including Beirut the capital. The Abou Ali River provides water to the districts of Bcharre, Zgharta, Minieh, and Tripoli, the second-largest city in the country. The Ibrahim River emerges from the Afqa grotto and supplies water for the districts of Kesrouan and Jbeil, a touristic area on the coast including the Byblos archaeological site. The El Kelb River basin supply water for the El Metn and Keserouan districts. The Jeita spring in the lower area of the El Kelb river basin provides 75% of the domestic water supply for the Beirut district and its northern suburbs (about 1.5 million inhabitants) (Doummar et al., 2014). The contribution of the snowfed upper cretaceous plateau to the discharge at the Jeita spring (60 m a.s.l.) was estimated at 39% (Margane et al., 2013). All three basins have similar topographic, physiographic and geological characteristics. They stretch from the Mediterranean Sea to the Mount Lebanon with a mean west-facing slope. The recharge area on Mount Lebanon above 1200 m a.s.l. can be classified as mid- to high-altitude ranges with rugged topography and high plateaus (Viviroli et al., 2007). The maximum elevation exceeds 2500 m a.s.l. in the three basins. The Abou Ali river basin includes Qurnat El Sawda (3088 m a.s.l.) the highest point in Lebanon and the Levant. The land cover is mainly bare rocks and soils and short scrublands.

The surface drainage network is poorly developed in the upper areas of the basins as most rainfall and snowmelt-runoff infiltrates in the highly karstified limestone aquifers and aquitards beneath the surface of Mount Lebanon (Margane et al., 2013). Basin outlines used in this study (Figure 1; Table 1) were derived from surface topography and may differ from the actual catchments boundaries which are mostly controlled by the karst aquifer geometry and connectivity (Margane et al.,





2013). The topographic catchments are shown here to give an indication of the basin size and location because the hydrogeological catchments are poorly known. Meteorological data at the Beirut weather station (20 m a.s.l.) operated by the national civil aviation service for the time period between 1984 and 2016 suggest that W1314 was a very dry year (fifth driest year on record between 1984 and 2016). W1415 and W1516 were less dry but had below average precipitation (Table 2).

**Table 1.** main characteristics of the river basins in this study (defined using the surface topography). Annual flow based on unpublished data from the Litani river authority gauging stations (1992-2016).

| Basin name | Area, [km²] | Median elevation, [m a.s.l.] | Maximum elevation, [m a.s.l.] | Annual flow (standard deviation), [Million cubic meter per year] |
|---|---|---|---|---|
| Abou Ali | 513 | 1202 | 3088 | 210 (78.3) |
| Ibrahim | 323 | 1547 | 2681 | 321 (146.4) |
| El Kelb | 255 | 1381 | 2619 | 181 (85.9) |

**Table 2.** Precipitation anomalies in Beirut (1984-2016). Observed precipitation anomalies in Beirut (20 m a.s.l) are based on AWS
meteorological data provided by the national civil aviation service.

| Year | Observed precipitation anomalies in Beirut (1984-2016), [mm] | Climatological ranking of the observed precipitation in Beirut (1984-2016), driest = 1 and wettest = 33 |
|---|---|---|
| 2016 | -50.3 | 14 |
| 2015 | -45.8 | 15 |
| 2014 | -219.0 | 5 |



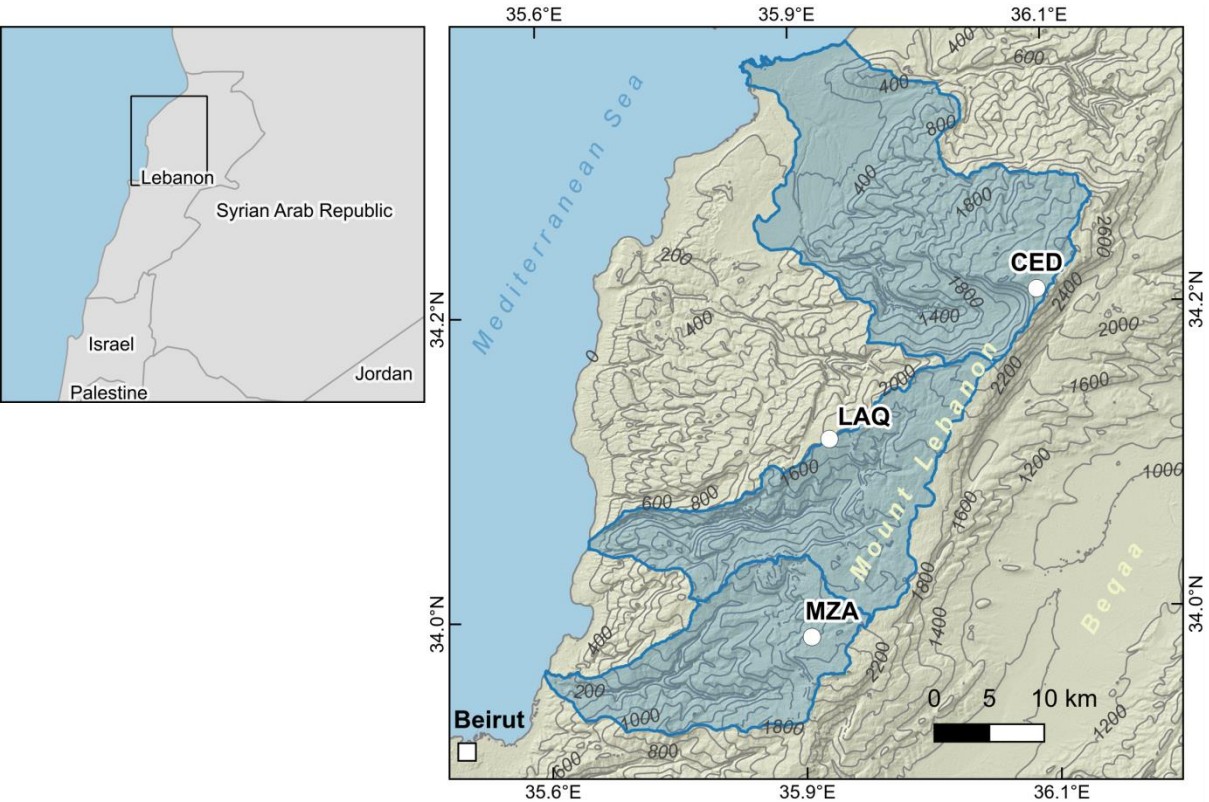

**Figure 1.** Study area. The three studied basins are marked in blue (from north to south: Abou Ali, Ibrahim, and El Kelb). Elevation contours are given in meters above the WGS-84 ellipsoid.

## 3 Methods

### 3.1 Model

We used SnowModel (Liston and Elder, 2006a), to simulate the spatial distribution of snow depth, SWE, and snowmelt over the study area. SnowModel is a physically-based snow evolution model which combines four submodels: (1) MicroMet (Liston and Elder, 2006b) spatially interpolates the meteorological forcing over the simulation domain from AWS observations and the digital elevation model (DEM) using the Barnes scheme; (2) EnBal, simulates the snowpack energy balance (Liston, 1995;

Liston et al., 1999); (3) SnowPack simulates the snow depth evolution and runoff (Liston and Elder, 2006b); and (4) SnowTran-3D simulates blowing snow sublimation and wind-driven redistribution processes (Liston et al., 2007). Two additional submodels are available but were not not used in this study: SnowAssim (Liston and Hiemstra, 2008) which allows the assimilation of SWE data and HydroFlow (Liston and Mernild, 2012), a runoff routing model. Pflug et al. (2019) implemented a gravity-driven scheme to compute the liquid water percolation through the snowpack instead of the default density threshold





scheme in SnowPack submodel. In the next sections we used the default model and this new version referred to as Pflug et al. (2019).

## 3.2 Model setup

The model was run at half-hourly time step over a simulation domain that encompasses the three study basins (Figure 1). The model grid resolution was set to 100 m, which is deemed sufficient to represent the main effects of the terrain on the energy
and mass balance (Baba et al., 2019). The input DEM was generated by cubic resampling of the ASTER GDEM V2 in the WGS-84 UTM 36N reference system. The land cover grid was obtained from the land cover and land-use map of Lebanon produced at a spatial resolution of 25 m (National Council for Scientific Research, 2015). Land classes were aggregated into 9 classes after Liston and Elder (2006a) and resampled to 100 m spatial resolution using a majority filter: Bare soils and rocks (59%), shrubland (12%), sea (8%), crops (7%), grassland (4%), clear-cut conifer (4%), urban (3%), coniferous, deciduous and
mixed forests (<3%), and water (<1%). The different land classes are needed to account for the snow holding capacities and canopy radiation effects (Liston and Elder, 2006a), however bare soil is the largely dominant class in the snow dominated region of the study area.

SnowModel was forced using the half-hourly meteorological data collected at the AWSs (LAQ, MZA, CED) for three snow seasons between 2013 and 2016 (01 November to 30 June). The data were taken from Fayad et al. (2017c) and only reformatted
to match SnowModel input format. We used the following input meteorological variables: precipitation (LAQ, MZA), air temperature (LAQ, MZA, CED), air relative humidity (LAQ, MZA, CED), wind speed and direction (LAQ, MZA, CED). No observations were available for LAQ AWS during snow season 2013-2014 (Fayad et al., 2017b). We did not use the shortwave radiation measurements because of the large data gaps. The model was used with all default parameters with two exceptions. First, the snow-rain threshold was set to 0°C instead of 2°C based on the analysis of the local meteorological record and other
studies in similar climatic context (Harpold et al., 2017). Second, we did not activate the default correction of the precipitation rate with elevation. It means that we ran the model with the minimal hypothesis that the precipitation spatial distribution is only controlled by the distances with the AWS according to the Barnes interpolation scheme (Liston et al., 2006b). This approximation is justified by the good horizontal (Figure 1) and vertical distribution (Figure 2) of the AWS in the simulation domain and the fact that we could not identify a robust elevation effect from the AWS precipitation records.
The liquid water percolation scheme by Pflug et al. (2019) introduces an additional parameter, a freezing curve parameter after Jordan, (1991), that we left to the default value of 50 kg m$^{-3}$ following Pflug et al. (2019). The gravity-drainage scheme implies to run the simulation with the multi-layer snowpack option that is not activated in the default run.





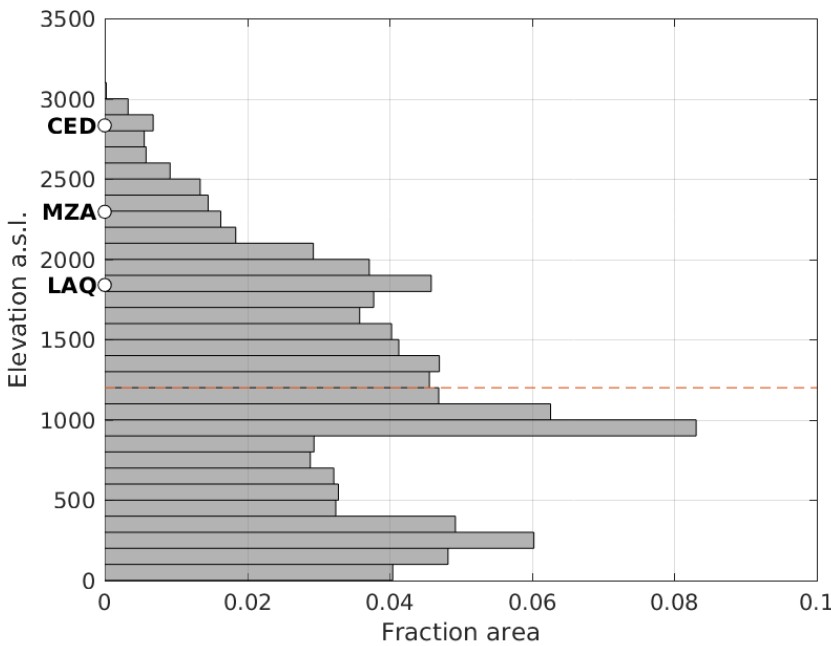

**Figure 2.** Elevation histogram of the simulation domain (in meters above mean sea level) including the position of the AWS. The dashed line indicates the reference elevation which is generally considered as the lower winter snowline elevation in Mount Lebanon, i.e. the elevation below which the snow cover is virtually absent (1200 m).

### 3.3 Model evaluation

The validation dataset includes (1) half-hourly snow height (HS), snow albedo and incoming shortwave radiation collected at
the AWS from W1314 to W1516, (2) bi-weekly manual snow density measurements collected near the AWS during the snow seasons of W1415 and W1516, and (3) daily snow cover area (SCA) observations from MODIS from W1314 to W1516. All these data are fully described in Fayad et al. (2017b) and are available as open data in a public repository (Fayad et al., 2017c).

Nearly continuous snow depth is available for water years W1415 and W1516 at LAQ and for water years W1314, W1415 and W1516 at MZA and CED. Half-hourly snow depth records were averaged to the daily timestep. We computed the daily
albedo of the surface below AWS from half-hourly upward and downward-looking pyranometers measurements following Stroeve et al. (2013), i.e. by computing the ratio of the daily incoming and reflected shortwave radiation totals. Summer albedo values ranged between 0.2 and 0.4 depending on the site and the year, hence we removed albedo values lower than 0.5 to keep only snow albedo measurements. The calculated daily snow albedo values and the incoming shortwave radiation were averaged to monthly values. The MODIS dataset is a daily cloud-free time series of snow cover maps providing the snow presence and
absence at 500 m resolution (binary SCA). The model outputs at 100 m resolution were converted to binary SCA using a threshold of 10 mm w.e. of SWE.





### 3.3 SWE estimation

We used the model outputs to estimate the evolution of SWE over the three basins. The daily distributed SWE was spatially integrated over the area of each basin located above 1200 m a.s.l.. From the temporal evolution of these basin-scale SWE time
series we derived the following key indicators: date and value of the peak SWE (maximum SWE during a water year), snow melt-out date (first day of the calendar year on which SWE gets below 1 mm w.e.).

### 4 Results

### 4.1 Model evaluation

Figure 3 compares the observed and modelled SWE evolution using both SnowModel configurations. The Pflug et al. (2019)
model provides a better simulation of SWE during the melt season in CED and MZA. At LAQ, both models are positively biased.

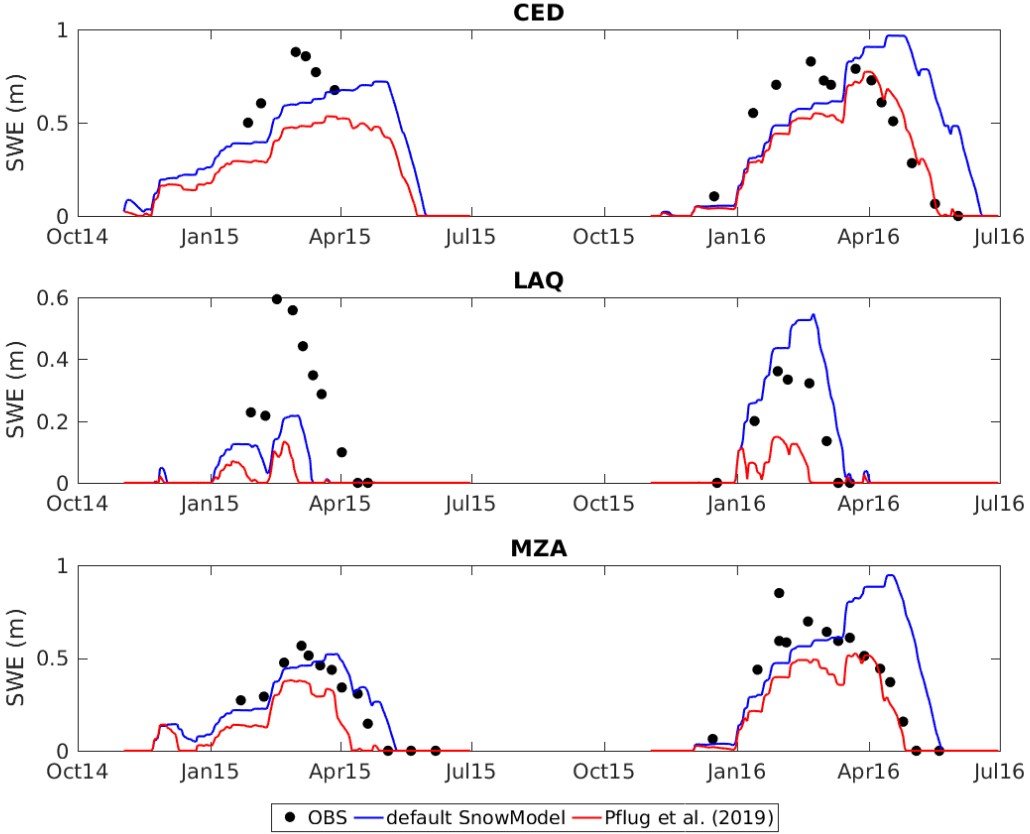

**Figure 3.** Time series of observed and modelled daily SWE at each AWS. SWE data were obtained from manual sampling during snow course measurements.





Figure 4 compares the modelled snow depth time series with the continuous snow depth measurements at each AWS. Although the difference between both models are less marked, similar observations as above can be made. This comparison also shows that the model reproduces well the snow depth evolution during W1314 and W1415 at CED and MZA, and the Pflug et al. (2019) model seems to be more consistent at these stations. The model overestimates snow depth at MZA during W1516 but the snow depth measurements are very noisy during this period suggesting that the data may be affected by a sensor issue,

especially given the SWE observations presented above. Otherwise, the same positive bias as noted above (Figure 3) can be observed at LAQ for W1415.

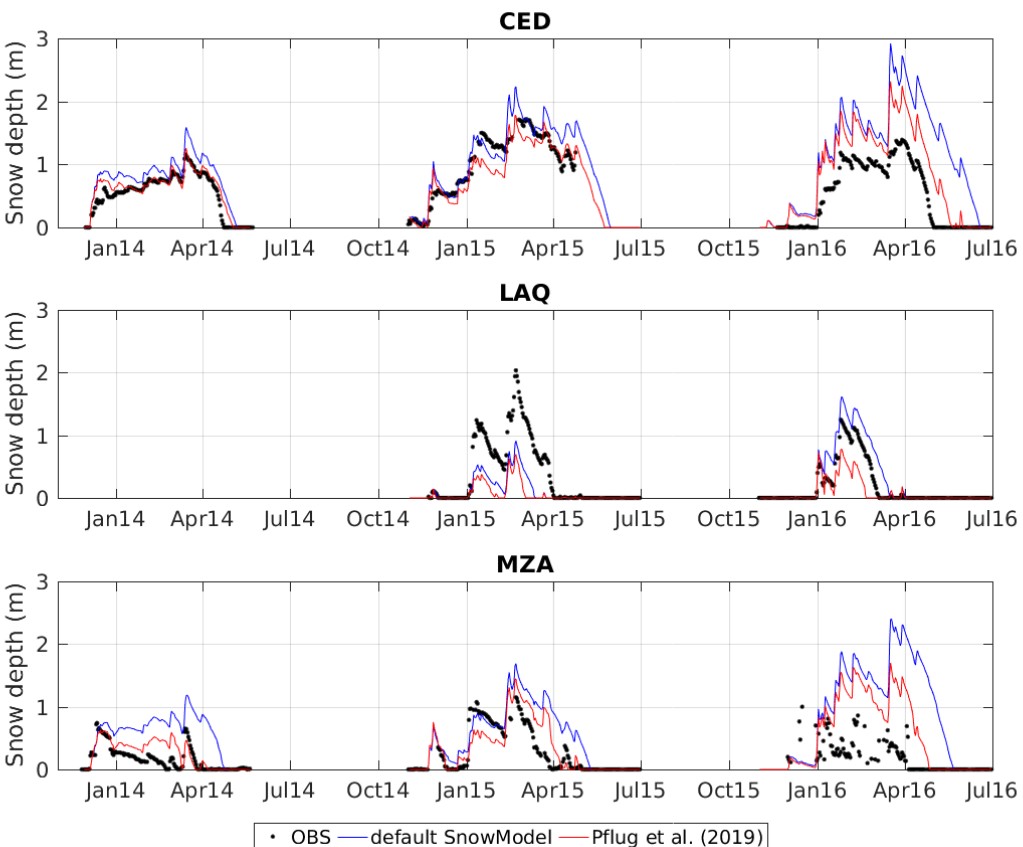

**Figure 4.** Time series of observed and modelled daily snow depth at each AWS.

The comparison of the modelled and observed snow cover shows that both models perform well in reproducing the snow cover evolution at the catchment scale, although the default model tend to overestimate the snow cover area during the ablation periods. This is particularly evident in spring 2016, but also during the atypical melt event in January 2014.

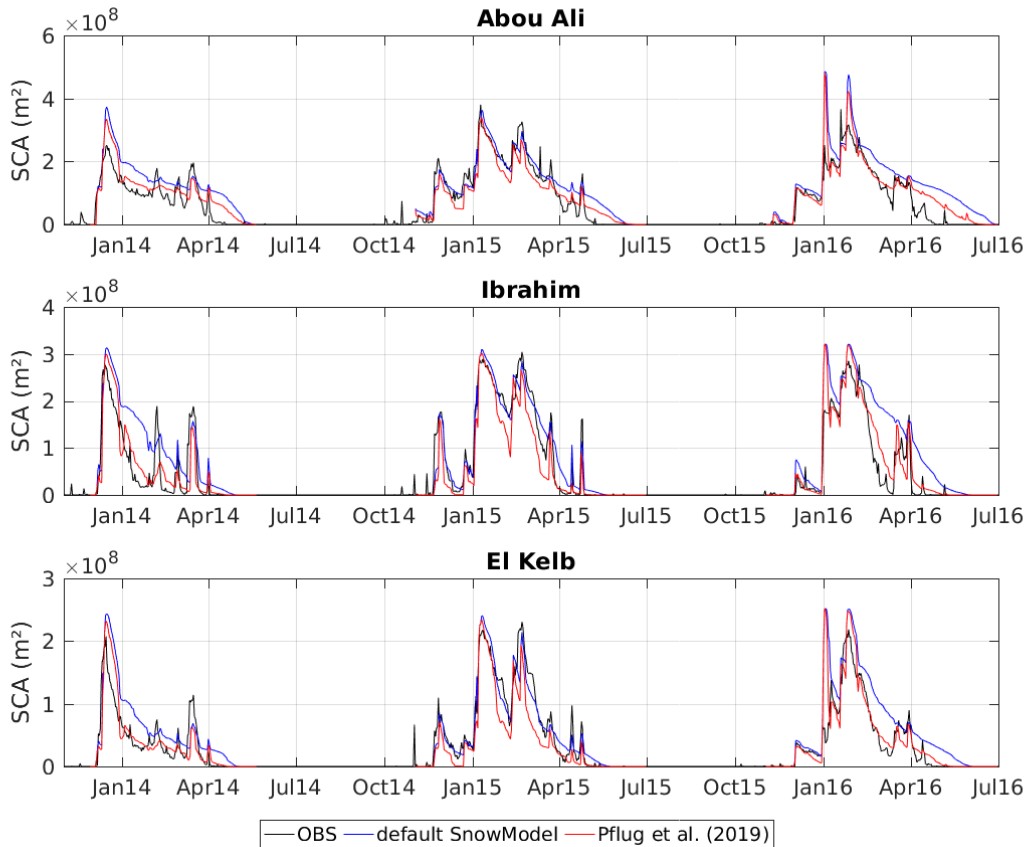

**Figure 5.** Time series of observed and modelled daily snow cover area in each study catchment.


Overall, Figures 3, 4 and 5 suggest that the Pflug et al., (2019) model provides a better representation of the snow cover in terms of snow depth, SWE and SCA. Table 3 shows that the correlation and RMSE are generally improved in comparison with the default model. The notable exceptions are small increases in the RMSE of snow depth and SWE at LAQ, which are hardly conclusive given the large accumulation biases at this AWS (see above, Figure 3 and Figure 4). At CED and MZA, the

performances of the model are significantly improved with the Pflug et al. (2019) version in terms of both correlation and RMSE with observed SWE. Given the better performance of the Pflug et al. (2019) model in this study area, we use this model in the following sections.

Figure 6 compares the monthly observed and modelled shortwave radiation and snow albedo at the three AWS. The incoming shortwave is well simulated with correlation coefficients ranging between 0.94 and 0.99 and RMSE ranging between

13 W m$^{-2}$ (LAQ) and 32 W m$^{-2}$ (MZA). The snow albedo variability is less accurately captured but the RMSE remains acceptable with 0.086, 0.074, and 0.096 at CED, LAQ, and MZA respectively.





**Table 3.** Performance of both model runs, default SnowModel and Pflug et al. (2019). Where, SD is daily snow depth from continuous
acoustic gauges, SWE is snow water equivalent from snow course measurements, and SCA is daily snow cover area from MODIS. The
value in bold indicate the best performance.

|  |  | Correlation (r) |  | RMSE |  |
|  |  | Default | Pflug | Default | Pflug |
| --- | --- | --- | --- | --- | --- |
| SD, [m] | CED | 0.73 | **0.84** | 0.67 | **0.38** |
|  | LAQ | 0.72 | **0.76** | **0.32** | 0.38 |
|  | MZA | 0.41 | **0.58** | 0.79 | **0.45** |
| SWE, [m] | CED | 0.28 | **0.78** | 0.30 | **0.22** |
|  | LAQ | 0.39 | **0.56** | **0.21** | 0.26 |
|  | MZA | 0.58 | **0.93** | 0.25 | **0.16** |
| SCA, [km$^{-2}$] | Abou Ali | 0.89 | **0.91** | 56 | **40** |
|  | Ibrahim | 0.90 | **0.93** | 49 | **36** |
|  | El Kelb | **0.89** | 0.88 | 32 | **30** |
| **Best model count** | | **1** | **8** | **2** | **7** |

Finally, we evaluate the model using the snow cover duration from MODIS over the entire study area (Figure 7). The results
show that the spatial patterns of snow cover duration are generally well reproduced. There is a positive bias in the northernmost
region, which corresponds to the region with the highest elevation (2700 to 3000 m a.s.l.). However, the average bias over the
entire domain remains low (2 days).





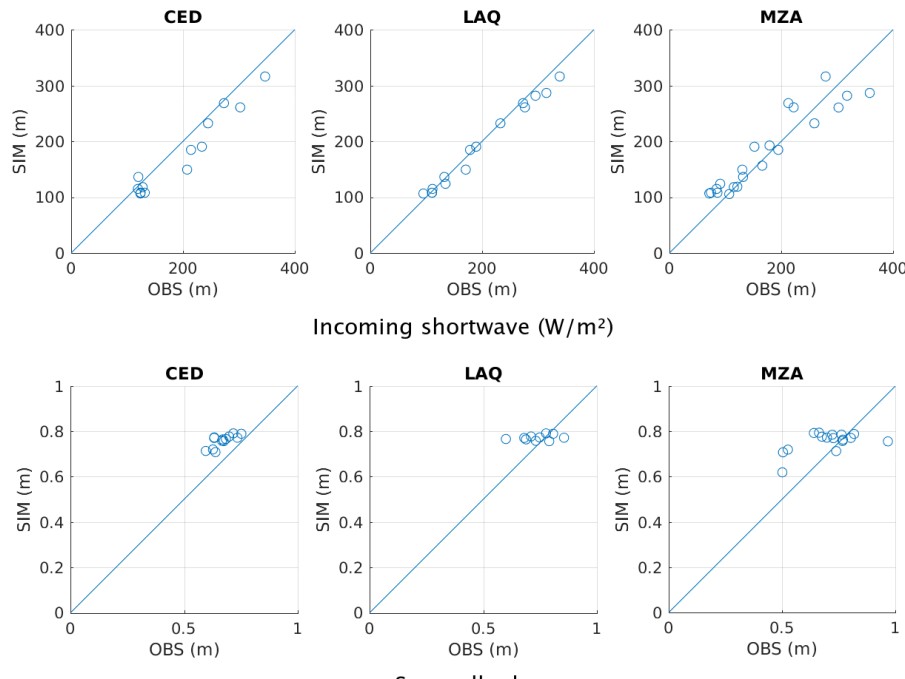

**Figure 6.** Scatterplots of observed vs. modelled monthly incoming shortwave radiation and snow albedo. Here the Pflug et al. (2019) model
was used.

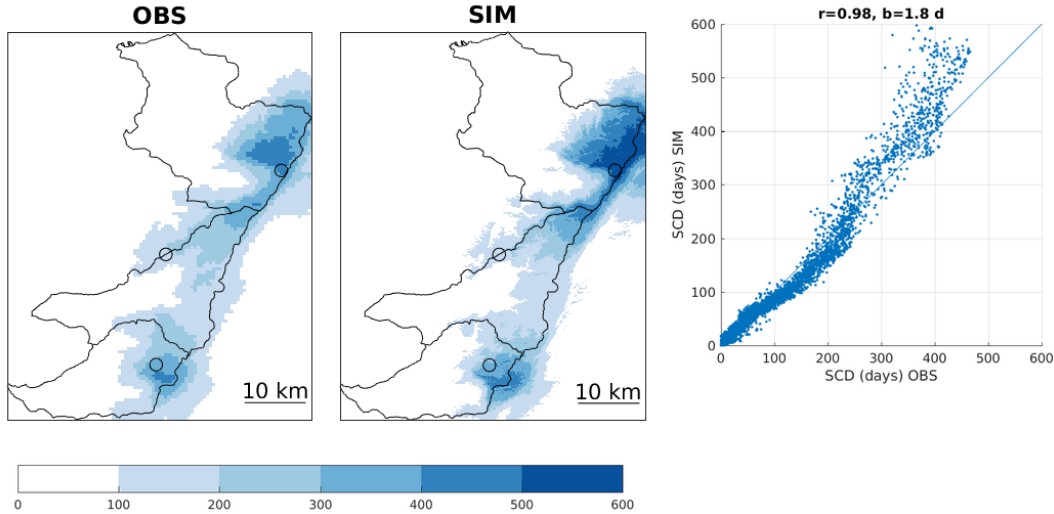

**Figure 7.** Map of the snow cover duration (SCD in days),computed over the entire period of the simulation (three snow seasons from 01
November 2013 to 01 July 2016), using the Pflug et al. (2019) model. The black circle indicates the location of the AWS (Figure 1). The
scatterplot on the right shows the pixel-by-pixel comparison of the simulated and MODIS SCD (blue line is the 1:1 line, r is the correlation
and b the bias in days.).


## 4.2 Estimation of SWE

The evolution of the spatially-averaged modelled SWE over the study basins above 1200 m a.s.l. is shown in Figure 8 and the main indicators are provided in Table 4. Figure 8 shows that there is a large variability in the SWE evolution during the study period hence the peak SWE is not clearly defined. However, it was consistently reached on the same date in each catchment

230 for a given water year. During W1314 it occurred early in the snow season on 22 December 2013, then on 21 February 2015 and 27 January 2016. However, the maximum SWE was nearly reached on March 28 in the Abou Ali catchment, which comprises the highest elevation areas of the study area. The same Abou Ali catchment had the longest snow season, with a melt-out date ranging between 06 May and 08 June, while the melt-out occurred about a month earlier in El Kelb and Ibrahim catchments.

235  In terms of SWE values, 2015-2016 was the snow season with the largest peak SWE, followed by 2014-2015 and 2013-2014 in the three catchments (Table 4). This is also illustrated in Figure 9, which presents the temporal evolution of the total mass of snow (expressed in cubic meters of water equivalent) for each catchment. The Abou Ali has the largest snow mass throughout the snow season due to its larger area and higher elevation. The temporal evolution of the three catchments was similar in W1415 and W1516, however this figure also reveals that the behaviour of Abou Ali was different during W1314,

240 with a much larger snow mass in Abou Ali in spring than in El Kelb and Ibrahim catchments. The months with the largest mean snow mass in El Kelb and Ibrahim catchments were December in W1314, and February in W1415 and W1516, while in Abou Ali it was February in W1314, March in W1415 and February in W1516.

**Table 4.** Key indicators from the spatially-averaged SWE above 1200 m a.s.l. (also see Figure 10): date and value of the peak SWE and

245 melt-out date. Here the Pflug et al. (2019) results are reported.

| Basin | Peak SWE value, [mm] | | | Peak SWE date | | | Melt-out date | | |
|---|---|---|---|---|---|---|---|---|---|
| Water Year | W1314 | W1415 | W1516 | W1314 | W1415 | W1516 | W1314 | W1415 | W1516 |
| Abou Ali | 106 | 234 | 286 | 22-Dec | 21-Feb | 11-Feb | 07-May | 11-Jun | 09-Jun |
| Ibrahim | 73 | 155 | 202 | 22-Dec | 21-Feb | 11-Feb | 09-Apr | 21-Apr | 02-May |
| El Kelb | 73 | 141 | 186 | 22-Dec | 21-Feb | 29-Jan | 16-Apr | 02-May | 05-May |

Based on Figure 9, we selected the 01 March as a common date to represent the spatial distribution of SWE across the study domain for the three study years (Figure 10). These maps further illustrate how the SWE in Mount Lebanon can be strongly impacted by the interannual variability. On 01 March 2016, SWE exceeded 500 mm w.e. over a large portion of the Mount

250 Lebanon range, while it was below 200 mm w.e. on 01 March 2014.



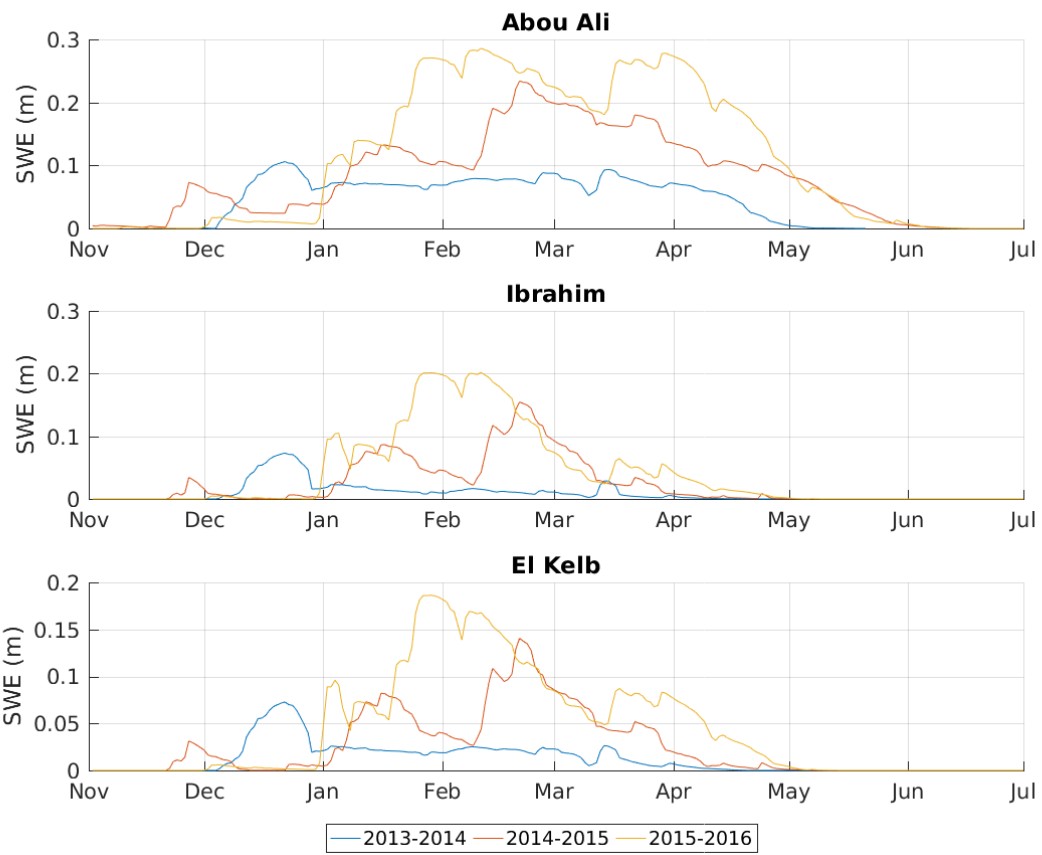

**Figure 8.** Temporal evolution of the simulated SWE above 1200 m a.s.l. (swed in mm), based on the Pflug et al. (2019) model outputs.

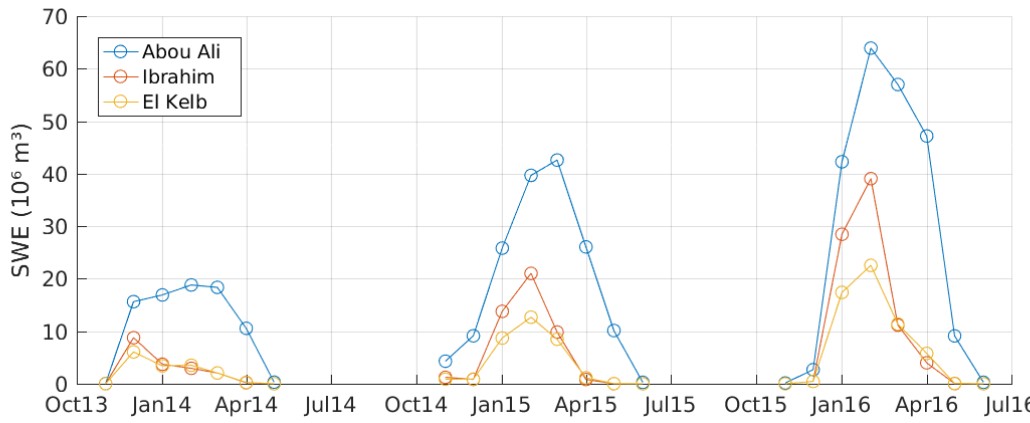

255

**Figure 9.** Temporal evolution of the spatially-integrated mean monthly SWE in cubic meters (total snow mass in each catchment), based on the Pflug et al. (2019) model outputs.



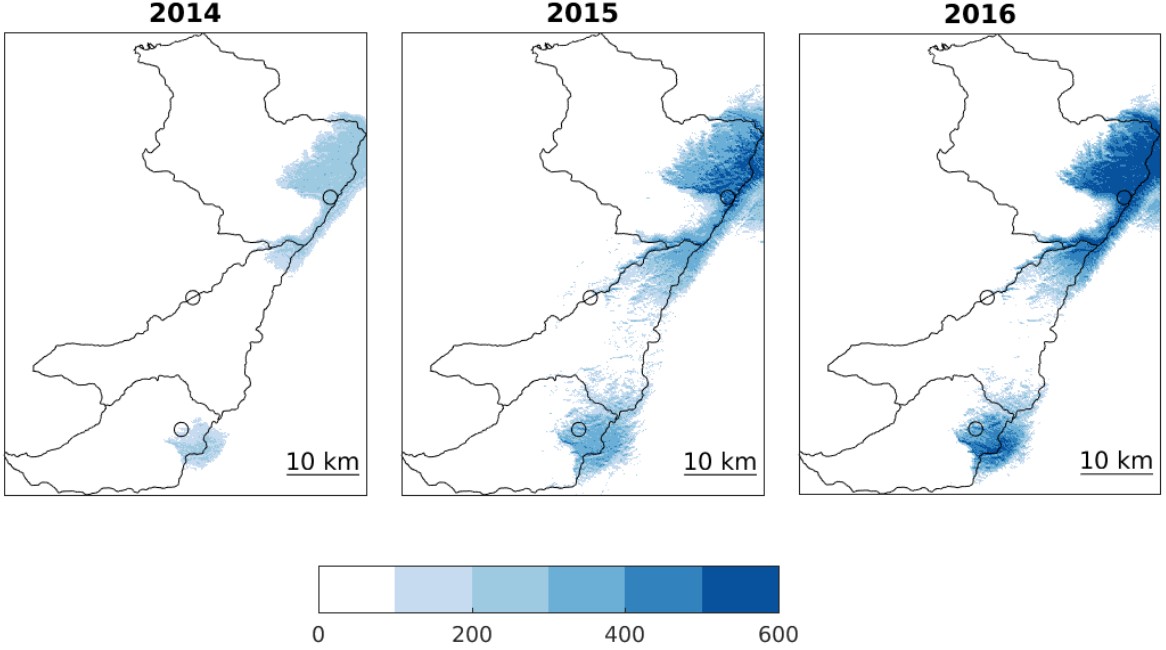

**Figure 10.** Spatial distribution of the simulated SWE (in mm) on 01 March, here the Pflug et al. (2019) model outputs are shown.

## 5 Discussion

Previous studies showed that SnowModel can overestimate SWE of warm maritime snowpack due to its liquid water percolation parameterization (Sproles et al., 2013; Pflug et al., 2019). We also find that the default SnowModel version overestimates SWE at MZA and CED stations. The available observations show that new percolation scheme by Pflug et al. (2019) allows a clear improvement in modelled SWE but also in the snow depth and snow cover area at the catchment scale.

The improvement with the Pflug et al. (2019) model is similar to what has been reported by the same authors in the Olympic Mountains. The gravity drainage scheme allows a better representation of the liquid water percolation during the melt season, whereas the default density threshold method retains too much liquid water when applied to warm maritime snowpacks. Only at LAQ the Pflug et al. (2019) version does not improve the SWE simulation. However, the snowpack is shallower and the SWE error is already large during the accumulation period at this AWS hence it is difficult to consider this AWS as good reference for evaluating a liquid water percolation scheme. This error in SWE at LAQ is probably due to errors in the precipitation input data. This issue is being investigated but so far, we do not have a clear answer. In addition, the model tends to overestimate snow depth whereas there is no clear positive or negative bias in SWE. This suggests that the modelled density might be too low. High densification rates are typical of Mediterranean mountains (Fayad et al., 2017a) and this process might be insufficiently represented in SnowModel. Despite these limitations, however, the overall model evaluation with in situ and remote sensing data suggest that SnowModel modified by Pflug et al. (2019) generates a reasonable estimation of the SWE in





Mount Lebanon. It is not possible to make a direct comparison of the results of this study with previous snow studies since the periods of computation do not match, but we note that the orders of magnitude of the peak SWE are compatible (Sect. 1).

The agreement between the measured point scale and modelled SWE in the three basins indicated good agreement, with an r ranging between 0.56 and 0.93 and an RMSE ranging between 16 and 26 cm w.e. (Table 3). The range of the reported RMSE

values in Table 3 are not uncommon in Mediterranean like climates (Guan et al., 2013; Musselman et al., 2017). These results indicate that SnowModel modified by Pflug et al. (2019) was able to explain a large part (56-93%) of the variability in the SWE with an accuracy between 16 and 26 cm w.e..

Given the fact that the three AWS are well distributed in the domain (Figure 1), the reported RMSE values for SD, SWE, and SCA (Table 3) can be used as a possible estimate of the uncertainty of SWE in every grid cell. This uncertainty would be

a conservative estimate since the random part of the error in the distributed SWE is reduced by the spatial averaging. In addition, the MODIS data showed no evidence of a large systematic SWE error over the domain except in the highest elevation region of the Abou Ali catchment, where the model tends to overestimate the snow cover duration (Figure 7).

The model results tend to suggest that the spatial variability of SWE is rather low over Mount Lebanon, but this is due to the large-scale morphology of Mount Lebanon, with a strong topographic gradient from the coast to 2000 m a.s.l. followed by

a high elevation plateau. This plateau is situated above the zero degree isotherm in winter therefore the snowfall regime is rather homogeneous over this area. However, Mount Lebanon is also characterized by a peculiar heterogeneous topography, visible at smaller scales (below 100 m), with many karstic surface features like sinkholes, that create highly variable snow depth patterns due to wind-driven snow redistribution processes (Figure 11; Chakra et al. 2019). This variability is not resolved with this current model setup with a grid size of 100 m. This can be explained by the fact that SnowModel does not account

for subgrid variability and assumes spatial uniformity over the entire grid cell (Liston and Elder, 2006a). Not accounting for snow subgrid variability is considered a fundamental limitation in the application of snow models and have implications in solving for snow ablation (DeBeer and Pomeroy, 2017). Furthermore, the presence of heterogeneous fractional snow cover (Figure 11) have major implications on the surface energy fluxes (DeBeer and Pomeroy, 2009). If fact, the heterogeneity in snow cover fraction is known to have implications on the small-scale advection of sensible heat and other local variations in

the surface energy balance terms (DeBeer and Pomeroy, 2017). This interaction between the fractional snow cover and the overlaying boundary layer is usually simplified and not captured by most models (DeBeer and Pomeroy, 2009; Mott et al., 2018).

Figures 5 and 8 show multiple transient melt periods during the entire snow season. These events can be clearly observed in the three basins during the months of January and February (Figure 8), which is considered as the accumulation season.

Precipitation events during the same months did restore most of the snowpack (Figure 8). These events, which are more common in warmer climate, such as the Mediterranean climate of Mount Lebanon, make it more difficult to distinctly separate between the accumulation and ablation seasons (DeBeer and Pomeroy, 2017). These transient snowmelt and accumulation events pose major challenges in terms of simulating snow accumulation, its redistribution, and ablation due to the complex nature of these processes, which remain not fully understood (Liston, 1995; Essery et al., 2006).





Model results indicate that the largest snow reserves are found in the Abou Ali catchment (Figure 9). This is due to its larger area but also to a larger portion of the basin is located at higher elevation bands. As a result, the SWE evolution in the Abou Ali catchment differs from the El Kelb and Ibrahim catchment. The snow season is longer and the peak SWE occurs about a month later. Although the study period only spans three snow seasons, a large interannual variability can already be observed.

Both model simulations showed higher biases in computing SWE at lower elevation, where the default SnowModel tended
to overestimate and the Pflug et al. (2019) model tended to underestimate the SWE in LAQ (Figure 3). This can be attributed to the separation of rain from snow at lower elevations, and the fact that these regions are more sensitive to rain on snow events. Finally, the Pflug et al. (2019) snowpack liquid water percolation algorithm may be overestimating melt at lower elevations. To which extent these processes are influencing the model performance is still unknown for us.

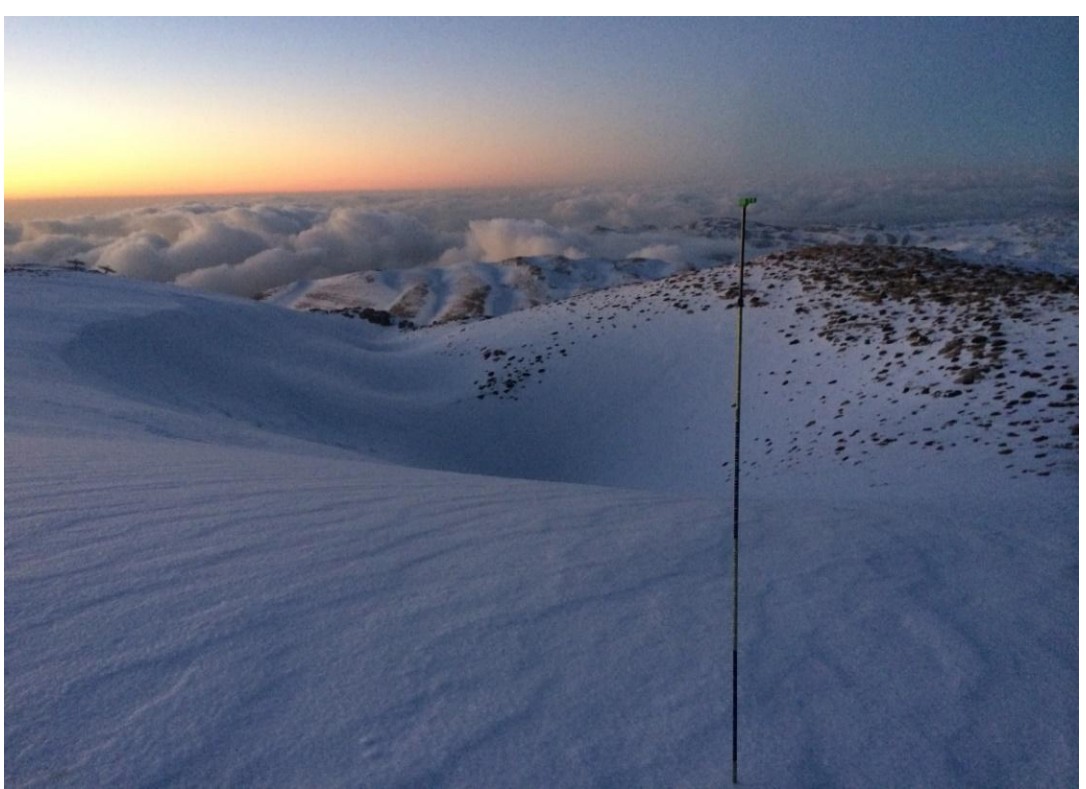


**Figure 11.** Snow redistribution following major snowfall event at 2310 m a.s.l. and 200 m away from the MZA AWS (located to the left of the image). Image taken on January 15th, 2016. Snow depth at the station was 7.5 cm. Snow depth measured with snow probes in the vicinity ranged between 0 and 258 cm (n = 46) with an average of 117 cm (median = 99 cm) and a standard deviation of 67 cm.



## 6 Conclusions

The snow cover is a critical yet poorly known water resource in many Mediterranean regions (Fayad et al., 2017a). A good knowledge of the snow water equivalent (SWE) and snowmelt in Mount Lebanon is key for the evaluation and management of the water resources in Lebanon. This study presents the first simulation of the daily SWE in Mount Lebanon over three contrasted snow seasons (2013-2016) in three catchments which are critical in terms of water resources for Lebanon. This was made possible by the availability of new meteorological AWS observations at high elevation (1840-2834 m a.s.l.). The model

was evaluated using a number of in situ and remote sensing observations. Recent developments with SnowModel including a gravity drainage formulation instead of the default snow density threshold to control liquid fluxes (Pflug et al., 2019) were beneficial in this study area where the snowpack is subject to multiple snowmelt episodes during the snow season. The results indicated that this improved model was generally able to reproduce the observed snowpack properties with a reasonable performance despite the uncertainties in the model forcing. Although the default version of SnowModel can theoretically be

applied in any climatic region since it relies mostly on physical equations to solve the energy and mass balance of the snowpack, it also includes parameterizations to replace processes that are too complex to be physically represented in the model. These parameterizations were primarily derived from studies in Arctic or subarctic regions (e.g. Bruland et al., 2004; Mernild et al., 2008) and should be carefully examined if the model is applied to Mediterranean mountain regions with mild, humid winters.

Over the simulation period 2013-2016, the maximum snow mass was reached between December and March and its magnitude changed significantly from year to year in the three study catchments. The results are subject to large uncertainties given that the model was only partially evaluated despite all the efforts made to collect in situ measurements. In particular there is a mismatch between the spatial scale of field data and the current model extent and resolution which prevents a robust estimation of the model error. To further evaluate the model ability to resolve small-scale variability, higher resolution remote

sensing products like Sentinel-2 SCA (Lebanon has been recently included in Theia Snow collection, Gascoin et al., 2019) or stereo-satellite snow depth (Marti et al., 2016) should be useful. Another important limitation of this work is the short record of the forcing data from the AWS that does not allow drawing robust conclusions on the interannual variability of the SWE in Mount Lebanon and even less on the extremes. In particular, exceptional snowfall occurred in Lebanon during winter 2018-2019 and the snow cover was still significant in July near MZA and CED stations. Future work may focus on the application

of downscaled climate reanalysis data to extend the study period (Mernild et al., 2017; Baba et al., 2018). Longer simulations would also enable to use the simulated snowmelt as input to a hydrogeological model of the karst to improve our knowledge on the snowpack dynamics influence on the spring discharges and regional water resources.

    We found in Mount Lebanon snow some typical features of Mediterranean snow, i.e. the high temporal variability and the high densification rates (Fayad et al., 2017a). Results from this study indicate that SnowModel modified by Pflug et al. (2019)

can provide realistic estimates of SWE, which are essential for understanding the hydrological contribution of the snowpack from Mount Lebanon. Refining the model grid to better account for wind redistribution of snow could improve the model

performance in capturing SWE. Additional processes are known to affect the snowpack dynamics in Mount Lebanon like rain-on-snow events and mineral dust deposition from the African Sahara and the Arabian Desert. Dust radiative forcing is not included in SnowModel, but recent studies in regions with similar climate suggest that it can have a significant impact on

snowmelt runoff (e.g. Painter et al., 2018). Finally, a specific feature of Mount Lebanon is that most of the snow accumulates on a high elevation plateau. This morphology implies that the seasonal snow cover could be highly impacted by a rise in the mean zero-degree isotherm altitude in winter, which determines the partition of precipitation into rain or snow. Given the importance of snowmelt for groundwater recharge and water supply in Lebanon it is urgent to evaluate the vulnerability of the snowfall regime to climate change in Mount Lebanon.

**Code and data availability**

The SnowModel code is archived at https://github.com/jupflug/SnowModel.
Model forcing and validation data are available at https://doi.org/10.5281/zenodo.583733.

**Author Contributions**

Both A.F. and S.G. worked on the conceptualization of the research goals and the design of the methodology as well as
analyzing the model outputs. A.F. and S.G. contributed equally in the writing and editing of the manuscript.

**Acknowledgement**

This publication was made possible through support provided by the French National Research Institute for Sustainable Development (IRD) and the Centre national d'études spatiales (CNES) via its Tosca program. Part of this work has been carried out within the frame of the ERANET-MED03-62 CHAAMS project.

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
