# Peer review of "The role of liquid water percolation representation to estimate snow water equivalent in a Mediterranean mountain region (Mount Lebanon)"

_Hydrology and Earth System Sciences, 2019_

## Referee Comment (RC1) · Anonymous Referee #1 · 5 Nov 2019

This paper presents the application of SnowModel for the Lebanon mountains using the traditional configuration, and a new liquid water percolation into the snow that permits improve the calculations of snow depth and snow density over the study area. The paper is interesting first for showing that liquid water percolation has a major implications in snow modeling; but also to see how with very limited data it is possible to obtain a reasonable good distributed representation of snow in areas where very little information was available. This result itself justifies the publication of the paper in HESS.

[Figure]

In my opinion, the paper is convincing when demonstrating that the simulations made with new scheme for water percolation produces better simulations than the traditional configuration. The only limitation (that is fully understable) is the lack of field data to assess that the water percolation is better simulated actually. In other words, are the better results the consequence of better representing the physical processes within the snowpack, or is just because it just provides less SWE that is closer to observations?. I realize that is not easy to demonstrate this, but perhaps if authors show when the two simulations really differ in the temporal series (may be showing the accumulated differences of both simulations), and at that time percolation plays a major role it could point out that there is a real causal effect. It would be also nice to see the differences in the distributed snow duration maps using SnowModel under the two compared model configurations, it may also reveal some interesting finding to see which areas are more benefited from the new percolation model.

Another question, that is out of the scope of this paper but it could be just briefly discussed, is how much room there is for improving the simulations in the area. For instance, authors mention the importance of the determination of liquid/solid phase of precipitation. However, if I understand well it is used a very simple temperature threshold approach, when now there are much more sophisticated approaches. I also would like to know more about the improvement (or limitations) of the snow blowing and redistribution module used in the model. Does it really help to improve the spatial distribution of snow over the area?. Finally, I also guess that sublimation is another important component of the SEB in Lebanon (as in other Mediterranean Mountains), what does the model inform about this process, is it an important source of uncertainty for snow modeling in this area?.

I have not more significant comments about the manuscript. It is well written and structured and very easy to be followed by readers. Figures are simple and nice.

[Figure]

514, 2019.

---

## Referee Comment (RC2) · A.N. Arslan (Referee) · 12 Nov 2019

**Review comments** on hess-2019-514, entitled, "The role of liquid water percolation representation to estimate snow water equivalent in a Mediterranean mountain region (Mount Lebanon)".

The paper investigates the spatial distribution and evolution of the snow water equivalent (SWE) during three snow seasons (2013-2016) in the coastal mountains of Lebanon. A recent upgrade of the liquid water percolation scheme in SnowModel, which was introduced to improve the simulation of the snow water equivalent (SWE) and runoff in warm maritime regions was evaluated. The performance of the model was evaluated against continuous snow depth and snow albedo observations at the AWS, manual SWE measurements, and MODIS snow cover area.

Chapter 3.3: SWE estimation is very important as the main focus of this paper. But this chapter written very shortly. It would be good to make it more detailed like "how to estimate the evolution of SWE over the three basins were done using the model outputs and etc.?

How manual SWE measurements are conducted, what instruments are used?

In chapter 4.1: It is said that "Figure 3 compares the observed and modelled SWE evolution using both SnowModel configurations. The Pflug et al. (2019) model provides a better simulation of SWE during the melt season in CED and MZA. At LAQ, both models are positively 175 biased." I am not sure this is totally true. It seems that it is true for year 2016 but not for 2015!!! It is important to discuss this why it is like that any reasons? Why model works better for 2016 not for 2015?

What is the purpose of comparing snow depth measurements in Figure 4? This should be explained. Same thing is valid for SCA. What is the purpose of this comparison in terms of estimate SWE in this paper?

In generally I found this paper focuses on comparison of the performances of default model and the model with upgrade of the liquid water percolation scheme. Either title should be modified or focus should be more on the estimation of SWE in Mount Lebanon.

---

## Author Comment (AC1) · 1 Jan 2020

**Response to the reviewer #1**

We wish to thank the reviewer for the positive feedback on this submitted paper and the interesting comments. We are happy to follow the reviewer suggestions in a revised paper as detailed below.

Note: Reviewer's general comments are in "Black", reviewer's questions in **"Bold Black"** and authors comments in "Blue". Figures in the manuscript are referred by their 'Fig. number'. Revised figures are labeled '**Fig. R**'.

This paper presents the application of SnowModel for the Lebanon mountains using the traditional configuration, and a new liquid water percolation into the snow that permits improve the calculations of snow depth and snow density over the study area. The paper is interesting first for showing that liquid water percolation has a major implications in snow modeling; but also to see how with very limited data it is possible to obtain a reasonable good distributed representation of snow in areas where very little information was available. This result itself justifies the publication of the paper in HESS. In my opinion, the paper is convincing when demonstrating that the simulations made with new scheme for water percolation produces better simulations than the traditional configuration. **The only limitation (that is fully understable) is the lack of field data to assess that the water percolation is better simulated actually. In other words, are the better results the consequence of better representing the physical processes within the snowpack, or is just because it just provides less SWE that is closer to observations?** I realize that is not easy to demonstrate this, but perhaps if authors show when the two simulations really differ in the temporal series (may be showing the accumulated differences of both simulations), and at that time percolation plays a major role it could point out that there is a real causal effect.

We did not measure liquid water percolation in the snowpack during the field campaigns because we found out later that this aspect of the model is key to obtain good simulations. However, we can show the accumulated differences in time to illustrate when the liquid percolation scheme creates the differences in the simulated SWE. We propose to incorporate wet snow detection from Sentinel-1 data in the revised manuscript to better justify that we indeed get "the right answers for the right reasons" (Kirchner, 2006) despite the lack of liquid water in situ observations.

Figure R1. shows the time series of wet snow detection from Sentinel-1 over the period October 2014 to June 2016 (method below) and the simulated melt runoff for both model configurations. It can be observed that at CED and MZA, snowmelt runoff using the default model occurs in a short period of time when Sentinel-1 data suggest that the snowpack has completely melted (since dry snow is not possible during the spring season, the absence of wet snow means that there is no snow), as observed with the in situ SWE and HS data (Fig. 3

and Fig. 4, respectively). On the other hand, the runoff simulated using the Pflug et al. (2019) model is better synchronized with the wet snow occurrences, where wet snow occurs earlier in the season and is more temporally distributed. This is consistent with the expected behaviour of the new liquid percolation scheme, which allows a more gradual release of liquid water throughout the melt season when the snowpack is wet.

[Figure]

**Figure R1**: Time series of wet snow occurrences as detected from Sentinel-1 observations and modelled daily snowmelt runoff at each AWS.

**Method for detecting of wet snow from Sentinel-1 SAR**

The wet snow detection was done using Sentinel-1 SAR observations over the period October 2014 to June 2016. We extracted Sentinel-1 backscatter in VV polarization mode from the Sentinel-1 Ground Range Detected (GRD) collection in Google Earth Engine (Gorelick et al., 2017). We spatially averaged the backscatter at each station using a buffer with a radius of 100 m. Then, we defined a reference "dry" surface backscatter for each station using the 10th percentile of the backscatter time series (CED-8.4 dB, LAQ: -10.7 dB, MZA: -6.10 dB). A negative departure of 4 dB to this reference was used to determine the occurrences of wet snow (Nagler et al., 2016).

**It would be also nice to see the differences in the distributed snow duration maps using SnowModel under the two compared model configurations, it may also reveal some interesting finding to see which areas are more benefited from the new percolation model**.

We thank the reviewer for this suggestion, which indeed reveals that the Pflug et al. (2019) liquid water percolation scheme had a greater impact in the mid-elevation zones near 2000 m asl. This is consistent with the expected behaviour of this scheme since these areas are prone to continuous melting during the winter. The Pflug et al. (2019) scheme is expected to have a greater impact in areas where the snowpack is wet and isothermal.

[Figure]

**Figure R2**. Left: map of the difference in days between the snow cover duration (SCD) simulated by the default model and the Pflug et al. (2019) model. Right: mean difference by elevation band. The SCD was computed over the simulation period (three snow seasons from 01 November 2013 to 01 July 2016).

[Figure]

**Figure R3**. Top: maps of the difference in days between the simulated snow cover duration and the observed snow cover duration from MODIS. Bottom: scatterplots The SCD was computed over the simulation period (three snow seasons from 01 November 2013 to 01 July 2016).

In the revised manuscript we propose to replace Fig. 7 with Figure R3 and add Figure R2 to better discuss the spatial impact of the new liquid water percolation scheme.

Another question, that is out of the scope of this paper but it could be just briefly discussed, is **how much room there is for improving the simulations in the area.**

There is certainly room for improving the simulations.

1) The comparison with MODIS data suggests that the enhanced model performed better at mid-elevations but there remains a positive bias in the snow cover duration at high elevation (Fig. R3). In this study, the model was forced with observations collected from three AWS but precipitation data were not available at the highest AWS (CED). Since then a precipitation gauge was installed at CED and high elevation precipitation data are available. Therefore we suggest that the simulations could be improved by using observed precipitation from all stations to have a better representation of the high elevation precipitation volumes.

2) Given that temperatures remain close to 0°C during winter precipitation events over large areas of Mount Lebanon, a better parameterization of the precipitation phase partitioning is expected to enhance the simulations and help in better capturing the rain on snow events. In general, this is probably a key issue for Mediterranean mountain regions with mild winters (see comment below).

3) As discussed in the manuscript better representation of wind redistribution processes is important in this region. Extremely high variability of snow depth (few meters of difference over a distance of a hundred meters) could be seen over shorter distances (Figure 11, in manuscript). However, with limited information on the wind field and without a high resolution DEM it is difficult to assess blowing snow and its distribution at a finer scale. In addition, a proper representation of the snow redistribution could help in better explaining the role of advective heat fluxes.

4) Most of the energy used from snowmelt is shortwave radiation, with the occurrence of numerous dust storm events in this region, a proper parameterization of surface albedo, through assimilation of remote sensing products, as for example, could help in better capturing the onset of snowmelt especially at mid elevations. Collecting information on the radiative and thermal fluxes and measuring the properties of the different snowpack layers could help in better representing these processes in models.

In the revision, we can incorporate these insights into the Discussion section.

For instance, authors mention the **importance of the determination of liquid/solid phase of precipitation. However, if I understand well it is used a very simple temperature threshold approach, when now there are much more sophisticated approaches.**

The warmer nature of the Mediterranean climate of this study domain makes it challenging to set a proper threshold for the cutoff between rain and snowfall. Setting a proper static threshold for phase partitioning proved to be difficult, especially in the absence of local studies on the partitioning of precipitation in this area. We did use a modified snow-rain static temperature threshold after Harpold et al. (2017). We did not use a more complex precipitation-phase partitioning method such as those described in Harder and Pomeroy (2014) and Harpold et al. (2017) because the parameterization of such methods would require introducing more parameters that we will not be able to determine for this study area. In fact, having a better partitioning of the precipitation phase is one of the options to refining the model performance and improving the simulations in this area.

**I also would like to know more about the improvement (or limitations) of the snow blowing and redistribution module used in the model. Does it really help to improve the spatial distribution of snow over the area?.**

Blowing snow and its redistribution are important in this region. There is a high variability of snow depth at shorter distances as concluded from the field observations conducted by the

authors during two winter seasons. The existence of a large number of sinkholes in the region act as a trap for blowing snow  Figure 11. (in manuscript) is an example where snow depth varies from zero to more than 2.5 m over distances as short as a few hundred meters due to snow transport by wind. Previous work showed that the blowing snow and snow redistribution module in SnowModel improves the accuracy of the simulations (see Gascoin et al., 2013). Hence we chose not to focus on this aspect in this study and assume that it contributes to a realistic representation of the snowpack. We actually plan to work on this important aspect in a future study, but this would probably require to focus on a smaller model domain with an accurate fine scale elevation DEM that is currently not available in this region.

Finally, **I also guess that sublimation is another important component of the SEB in Lebanon (as in other Mediterranean Mountains), what does the model inform about this process, is it an important source of uncertainty for snow modeling in this area?.**

We agree that snow sublimation can be an important component of the snow mass balance. Figure R4 below shows the distribution of the mean annual sublimated snow in mm water equivalent. The sublimation reaches 70 mm we in the upper area of the study domain, and accounts for a fraction of about 5 to 15% of the annual peak SWE.

In SnowModel there two sources of sublimation: static surface sublimation and blowing snow sublimation. The relatively low sublimation rates are due to (1) high relative humidity due to the proximity to the Mediterranean Sea, which reduces both sources of sublimation (2) the high snow densification rates which inhibit blowing snow sublimation (Liston and Elder, 2006). We are unable to qualify the simulated sublimation however the point (2) is compatible with our field observations that the snow surface at the mid-elevation is subject to multiple melt and refreeze events. Such snowpack surface limits the capacity for snow removal by wind and hence reduces snow sublimation. To conclude the model suggests that the sublimation is probably not the main source of uncertainty in this area (precipitation remains the main uncertainty), but we suggest this aspect should be the focus of further investigation using in situ measurements by eddy covariance towers or lysimeters.

[Figure]

**Figure R4**. Mean annual snow sublimation in [mm] (Pflug et al. 2019 model).

I have not more significant comments about the manuscript. It is well written and structured and very easy to be followed by readers. Figures are simple and nice.

Again, we thank the reviewer for the kind feedback and insightful comments.

**References**

Gascoin, S., Lhermitte, S., Kinnard, C., Bortels, K., and Liston, G. E.: Wind effects on snow cover in Pascua-Lama, Dry Andes of Chile. Advances in Water Resources, 55, 25-39, 2013.

Gorelick, N., Hancher, M., Dixon, M., Ilyushchenko, S., Thau, D., and Moore, R.: Google Earth Engine: Planetary-scale geospatial analysis for everyone. Remote Sensing of Environment, 202, 18-27, 2017.

Harder, P., and Pomeroy, J. W. (2014). Hydrological model uncertainty due to precipitation-phase partitioning methods. Hydrological Processes, 28(14), 4311-4327, 2014.

Harpold, A.A., Kaplan, M.L., Klos, P.Z., Link, T., McNamara, J.P., Rajagopal, S., Schumer, R., and Steele, C.M.: Rain or snow: hydrologic processes, observations, prediction, and research needs, Hydrol. Earth Syst. Sci. 21, 1–22, 2017.

Kirchner, J.W.: Getting the right answers for the right reasons: Linking measurements, analyses, and models to advance the science of hydrology. Water Resources Research, 42(3), 2006.

Liston, G. E., and Elder, K.: A distributed snow-evolution modeling system (SnowModel). Journal of Hydrometeorology, 7(6), 1259-1276, 2006.

Nagler, T., Rott, H., Ripper, E., Bippus, G., and Hetzenecker, M.: Advancements for snowmelt monitoring by means of sentinel-1 SAR. Remote Sensing, 8(4), 348, 2016.

Pflug, J. M., Liston, G. E., Nijssen, B., and Lundquist, J. D.: Testing model representations of snowpack liquid water percolation across multiple climates. Water Resources Research, 55(6), 4820-4838, 2019.

---

## Author Comment (AC2) · 1 Jan 2020

**Response to the reviewer #2**

We wish to thank A.N. Arslan for the positive feedback on this submitted paper and the interesting comments. We are happy to follow the reviewer suggestions in a revised paper as detailed below.

Note: Reviewer's general comments are in "Black", reviewer's questions in **"Bold Black"** and authors comments in "Blue". Figures in the manuscript are referred by their 'Fig. number'. Revised figures are labeled '**Fig. R**'.

The paper investigates the spatial distribution and evolution of the snow water equivalent (SWE) during three snow seasons (2013-2016) in the coastal mountains of Lebanon. A recent upgrade of the liquid water percolation scheme in SnowModel, which was introduced to improve the simulation of the snow water equivalent (SWE) and runoff in warm maritime regions was evaluated. The performance of the model was evaluated against continuous snow depth and snow albedo observations at the AWS, manual SWE measurements, and MODIS snow cover area.

**Chapter 3.3: SWE estimation is very important as the main focus of this paper. But this chapter written very shortly. It would be good to make it more detailed like "how to estimate the evolution of SWE over the three basins were done using the model outputs and etc.?**

We thank the reviewer for this suggestion and we propose to update the section 3.3 as follows:

We used the model outputs to estimate the evolution of the SWE over the three basins. The basins surface boundaries were used to estimate the SWE contribution for each basin by computing the mean daily SWE for all pixels within each basin. To account for the hydrological contribution of SWE from the snow dominated regions, the daily distributed SWE was spatially integrated for elevations above 1200 m a.s.l..

From the temporal evolution of these basin-scale SWE time series we derived the following key indicators: date and value of the peak SWE (maximum SWE during a water year), snow melt-out date (the first day of the calendar year on which SWE gets below 1 mm w.e.), distributed SWE for the peak SWE data (this map is important from a hydrological perspective as it shows the amount of snow available for melt at the end of the snowfall season for each hydrological year).

We used the snow course locations to approximate the pixels that are representative for each snow course. The simulated SWE were evaluated by comparing the observed SWE (revisit time ~ 11 days on average for all snow courses according to Fayad et al. (2017b)) with the

simulated SWE value corresponding to the same dates when the in situ observations were taken. Five snow courses located near the three stations are used to showcase the evolution of SWE across different elevation bands (snow courses near the three AWS have elevations at 2823 and 2834 m a.s.l. for CED, 2297 and 2301 m a.s.l. for MZA, and 1843 m a.s.l. for LAQ).

**How manual SWE measurements are conducted, what instruments are used?**

A detailed description of the methodology and the protocols used for measuring SWE is described in Fayad et al. (2017b). We did not cover this in detail in this manuscript to minimize information redundancy.

**In chapter 4.1: It is said that "Figure 3 compares the observed and modelled SWE evolution using both SnowModel configurations. The Pflug et al. (2019) model provides a better simulation of SWE during the melt season in CED and MZA. At LAQ, both models are positively biased." I am not sure this is totally true. It seems that it is true for year 2016 but not for 2015!!! It is important to discuss this why it is like that any reasons? Why model works better for 2016 not for 2015?**

**We thank the reviewer for this comment. In the revised manuscript we propose to change the text as follows:** "At LAQ, both models are too biased to draw a robust conclusion on the effect of the percolation scheme. Both models are negatively biased in W1415 and either positively biased (default model) or negatively biased (Pflug et al., 2019) in W1516."

There are a number of variables that can influence the performance of the model such as blowing snow and snow redistribution at the pixel scale, snow course representativeness for each location, and precipitation observations used for forcing the model. This is why we present the results in Tab. 3 for the full simulation periods and all stations to make an assessment based on the overall model response. We think if is a strength of our study to have collected observations over three snow seasons at three AWS to reduce the effect of unexplained local uncertainties.

**What is the purpose of comparing snow depth measurements in Figure 4? This should be explained. Same thing is valid for SCA. What is the purpose of this comparison in terms of estimate SWE in this paper?**

The main reason for using HS and SCA in the validation of the SWE is attributed to the scarcity of the spatially distributed SWE data needed for model validation. In fact, HS is a good proxy of SWE in many cold regions climates (Strum et al., 2010). Furthermore, Fayad et al. (2017b) demonstrated that the estimation of SWE from HS could be achieved with acceptable accuracy using two years of observations in Mount Lebanon. Both, the SWE measurements that are available at a temporal resolution of ~ 10 days and the daily HS observations from

AWS are collected at point locations in Mnt. Lebanon (Fayad et al., 2017c) and only provide an idea on the evolution of the snowpack at that particular location.

The spatial distribution of HS, and to a similar extent the SWE, vary over short distances in Mediterranean Mountains (e.g. López-Moreno et al., 2011, 2013) and the spatial variability in snow depth is much greater than that of snow density (e.g. López-Moreno et al., 2013). The use of SCA is justified by the fact that extrapolating point SWE observations, to cover the high heterogeneity in mountain topography, is not possible given the limited number of point observations (~30 snow courses in total). Hence, in the absence of sufficient distributed SWE or HS, needed to validate the modeled distribution of SWE, the use of SCA as a proxy for SWE or HS was shown to be useful in similar mountain environments (e.g., Gascoin et al., 2015; Baba et al., 2018).

**In generally I found this paper focuses on comparison of the performances of default model and the model with upgrade of the liquid water percolation scheme. Either title should be modified or focus should be more on the estimation of SWE in Mount Lebanon.**

A previous version of this paper was focused on the SWE estimation but based on earlier discussions with the Editor we chose to change the focus on the finding that the percolation scheme has a significant impact on the simulation. We believe that this aspect can indeed be of interest to a broader audience. Yet, we use the "best" model configuration to compute the SWE in the final section as it is the societal question which motivated this study. The title "The role of liquid water percolation representation to estimate snow water equivalent in a Mediterranean mountain region (Mount Lebanon)" reflects these two aspects. We will do our best to clarify this double motivation in a revised manuscript, but we would like to keep this title if the Editor agrees.

**References**

Baba, M. W., Gascoin, S., Kinnard, C., Marchane, A., and Hanich, L.: Assimilation of Sentinel-2 Data into a Snowpack Model in the High Atlas of Morocco., Remote Sens, 10, 1982; doi:10.3390/rs10121982, 2018

Fayad, A., Gascoin, S., Faour, G., Fanise, P., and Drapeau, L.: Snow dataset for Mount-Lebanon (2011–2016), https://doi.org/10.5281/zenodo.583733, 2017c.

Fayad, A., Gascoin, S., Faour, G., Fanise, P., Drapeau, L., Somma, J., Fadel, A., Bitar, A.A. and Escadafal, R.: Snow observations in Mount Lebanon (2011–2016), Earth System Science Data, 9(2), 2017b.

Gascoin, S., Hagolle, O., Huc, M., Jarlan, L., Dejoux, J.-F., Szczypta, C., Marti, R., Sánchez, R.: A snow cover climatology for the Pyrenees from MODIS snow products, Hydrol. Earth Syst. Sci. 19, 2337–2351, 2015

López-Moreno, J.I., Fassnacht, S.R., Heath, J.T., Musselman, K.N., Revuelto, J., Latron, J., Morán-Tejeda, E., Jonas, T.: Small scale spatial variability of snow density and depth over

complex alpine terrain: Implications for estimating snow water equivalent. Adv. Water Resour. 55, 40–52, 2013.

López-Moreno, J.I., Fassnacht, S.R., Beguería, S., Latron, J.: Variability of snow depth at the plot scale: implications for mean depth estimation and sampling strategies. Cryosphere 5, 617–629, 2011.

Liston, G. E., and Elder, K.: A distributed snow-evolution modeling system (SnowModel). Journal of Hydrometeorology, 7(6), 1259-1276, 2006.

Pflug, J. M., Liston, G. E., Nijssen, B., and Lundquist, J. D.: Testing model representations of snowpack liquid water percolation across multiple climates. Water Resources Research, 55(6), 4820-4838, 2019.

Sturm, M., Taras, B., Liston, G., Derksen, C., Jonas, T., Lea, J.: Estimating snow water equivalent using snow depth data and climate classes. J. Hydrometeorol. 11, 1380–1394, 2010.

---

## Author Response (AR1)

**Reply to Editor R. Teuling**

The authors are thankful to R. Teuling for his positive comments on the revised version of this manuscript. We have updated the manuscript following R. Teuling recommendations and the comments and suggestions provided by the reviewers. Below are point-by-point response to the comments provided by the editor and the two reviewers. Editor/reviewers comments in **Bold** and the authors' responses are in blue. Changes in the manuscript are shown in track mode in the marked up version of the manuscript. We updated the numbering of figures and the reference section to reflect all changes made.

[R1] Authors mention the **importance of the determination of liquid/solid phase of precipitation. However, if I understand well it is used a very simple temperature threshold approach, when now there are much more sophisticated approaches.**

We added (lines 140 – 143): **"We did not use a more complex precipitation-phase partitioning method such as those described in Harder and Pomeroy (2014) and Harpold et al. (2017) because the parameterization of such methods**

**would require introducing more parameters that we will not be able to determine for this study area."**

[R1] **The only limitation (that is fully understandable) is the lack of field data to assess that the water percolation is better simulated actually. In other words, are the better results the consequence of better representing the physical**

**processes within the snowpack, or is just because it just provides less SWE that is closer to observations?**

[RT] **Please pay particular attention to the comment by referee #1 on the percolation, and consider including an additional figure on this process in the model simulations (even though a full validation might not be possible).**

We updated section 3.3 Model evaluation to include a proxy validation of the liquid water percolation using wet snow detection from sentinel-1. A description of the approach used is added at (lines 178 – 194): **"To further investigate the**

**effect of the liquid water percolation scheme on the internal properties of the snowpack, we used the wet snow detection from Sentinel-1 C-band Synthetic Aperture Radar data. We use these data as a proxy of melting conditions that we compared to the simulated release of liquid water from the snowpack. The wet snow detection was done using Sentinel-1 observations over the period October 2014 to June 2016. We extracted Sentinel-1 backscatter in VV polarization mode from the Sentinel-1 Ground Range Detected (GRD) collection in Google Earth Engine (Gorelick et**

**al., 2017). We spatially averaged the backscatter at each station using a buffer with a radius of 100 m. Then, we defined a reference "dry" surface backscatter for each station using the 10th percentile of the backscatter time series**

**(CED-8.4 dB, LAQ: -10.7 dB, MZA: -6.10 dB). A negative departure of 4 dB to this reference was used to determine the occurrences of wet snow (Nagler et al., 2016)."**

A figure (Figure 8) showing the wet snow occurrence and the modelled snowmelt runoff along with discussion were added under section 4.1 (lines 292 – 301):

**"Figure 8 shows the time series of wet snow detection from Sentinel-1 over the period October 2014 to June 2016 and the simulated melt runoff for both model configurations. It can be observed that at CED and MZA, snowmelt runoff using the default model occurs over a short period of time at the end of the snow melt season, while the runoff**

**simulated using the Pflug et al. (2019) model occurs earlier in the season and is more temporally distributed. This temporal distribution is consistent with Sentinel-1 data that indicate the presence of liquid water in the snowpack over the same periods. Again, this is consistent with the expected behaviour of the new liquid percolation scheme, which allows a more gradual release of liquid water throughout the melt season when the snowpack contains meltwater.**

[Figure]

**Figure 8: Time series of wet snow occurrences as detected from Sentinel-1 observations and modelled daily snowmelt runoff at each AWS.**

[R2] **SWE estimation is very important as the main focus of this paper. But this chapter written very shortly. It would be good to make it more detailed like "how to estimate the evolution of SWE over the three basins were done using the model outputs and etc.?**

We updated section 3.4 (lines 196 – 203) to read: **"We used the model outputs to estimate the evolution of the SWE over the three basins (Fig. 1). The basins surface boundaries were used to estimate the SWE contribution for each basin by**
**computing the mean daily SWE for all pixels within each basin. To account for the hydrological contribution of SWE from the snow dominated regions, the daily distributed SWE was spatially integrated for elevations above 1200 m a.s.l.. From the temporal evolution of these basin-scale SWE time series we derived the following key indicators: date and value of the peak SWE (maximum SWE during a water year), snow melt-out date (the first day of the calendar year on which SWE gets below 1 mm w.e.), distributed SWE for the peak SWE data (this map is important from a**
**hydrological perspective as it shows the amount of snow available for melt at the end of the snowfall season for each hydrological year)."**

We also updated the model evaluation section (section 3.3 lines 171 – 176) to highlight the use of SWE for model validation: **"We used the snow course locations to approximate the pixels that are representative for each snow course. The**
**simulated SWE were evaluated by comparing the observed SWE (revisit time ~ 11 days on average for all snow courses according to Fayad et al. (2017b)) with the simulated SWE value corresponding to the same dates when the in situ observations were taken. Five snow courses located near the three stations are used to showcase the evolution of SWE across different elevation bands (snow courses near the three AWS have elevations at 2823 and 2834 m a.s.l. for CED, 2297 and 2301 m a.s.l. for MZA, and 1843 m a.s.l. for LAQ)."**

[R2] **What is the purpose of comparing snow depth measurements in Figure 4? This should be explained. Same thing is valid for SCA. What is the purpose of this comparison in terms of estimate SWE in this paper?**

We added a paragraph to describe the importance of using SCA and HS in model validation (Section 3.3 model validation lines 188 – 194): **"The main reason for using daily HS, SCA, and Sentinel-1 wet snow observations in the validation of**
**the SWE is the scarcity of the spatially and temporally distributed SWE observations. However, HS is known to be a good proxy of SWE in many regions (Sturm et al., 2010). Furthermore, Fayad et al. (2017b) demonstrated that the estimation of SWE from HS could be achieved with acceptable accuracy using two years of observations in Mount Lebanon. Both the SWE and HS measurements are collected at point locations (Fayad et al., 2017c) and may not be representative at the catchment scale due to the high spatial variability of the snowpack in Mediterranean Mountains**

**(e.g. López-Moreno et al., 2011, 2013). The use of SCA is justified in the absence of distributed SWE or HS measurements to validate the model (e.g., Gascoin et al., 2015; Baba et al., 2018)."**

[R2] **"Figure 3 compares the observed and modelled SWE evolution using both SnowModel configurations. The Pflug**
**et al. (2019) model provides a better simulation of SWE during the melt season in CED and MZA. At LAQ, both models are positively biased." I am not sure this is totally true. It seems that it is true for year 2016 but not for 2015!!! It is important to discuss this why it is like that any reasons? Why model works better for 2016 not for 2015?**

We updated the model evaluation section (4.1) to address the difference between observed and simulated SWE. We added
(lines 212 – 214). **"At LAQ, both models are biased to draw a robust conclusion on the effect of the percolation scheme.**
**Both models are negatively biased in W1415 and either positively biased (default model) or negatively biased (Pflug et al., 2019) in W1516."**

And lines 304 – 309**: "Overall, all above results suggest that the Pflug et al., (2019) model provides a better representation of the snow cover in terms of snow depth, SWE and SCA. Table 3 shows that the correlation and RMSE are generally improved in comparison with the default model. The notable exceptions are small increases in the RMSE**
**of snow depth and SWE at LAQ, which are hardly conclusive given the large accumulation biases at this AWS (Fig. 3 and Fig. 4). At CED and MZA, the performances of the model are significantly improved with the Pflug et al. (2019) version in terms of both correlation and RMSE with observed SWE."**

[R1] **It would be also nice to see the differences in the distributed snow duration maps using SnowModel under the two compared model configurations, it may also reveal some interesting finding to see which areas are more benefited from**
**the new percolation model**.

The model evaluation section (4.1) Lines 236 – 289 where updated to reflect on the differences in the distributed snow duration maps using SnowModel under the two compared model configurations: **"We also evaluate the model using the snow cover duration from MODIS over the entire study area (Fig. 6). The results show that the spatial patterns of snow cover duration are generally better reproduced with the Pflug et al. (2019) scheme. The largest bias is found in the**
**northernmost region, which corresponds to the region with the highest elevation (2700 to 3000 m a.s.l.). However, the average bias for this model configuration over the entire domain remains low (2 days)."**

[Figure]

Figure 6. Top: difference in days between the simulated and MODIS SCD, computed over the entire period of the simulation (three snow seasons, from 01 November to 30 June, between 2013 and 2016) using the default SnowModel and the Pflug et al. (2019) model configurations. Bottom: the scatterplots show the pixel-by-pixel comparison of the simulated and MODIS SCD.

Interestingly, Figure 7 shows that the Pflug et al. (2019) scheme had a greater impact on the snow cover duration in the mid-elevation zones near 2000 m a.s.l. This is consistent with the expected behaviour of this scheme since these areas are prone to continuous melting during the winter and the Pflug et al. (2019) scheme is expected to have a greater impact in areas where the snowpack is wet and isothermal.

[Figure]

**Figure 7. Left: map of the difference in days between the snow cover duration (SCD) simulated by the default model and the Pflug et al. (2019) model. Right: mean difference by elevation band. The SCD was computed over the simulation period (three snow seasons (from 01 November 2013 to 01 July 2016).**

[R1] Another question, that is out of the scope of this paper but it could be just briefly discussed, **is how much room there is for improving the simulations in the area.**

We updated the conclusion (Section 6) to discuss some aspects for improving the model performance (lines 451 – 460 and lines 491 - 498).

[lines 451 – 460] **"The comparison with MODIS data suggests that the enhanced model performed better at mid-elevations but there remains a positive bias in the snow cover duration at high elevation (Fig. 6). In this study, the model was forced with observations collected from three AWS but precipitation data were not available at the highest AWS (CED). Since then a precipitation gauge was installed at CED and high elevation precipitation data are available. Therefore, we suggest that the simulations could be improved by using observed precipitation from all**

**stations to have a better representation of the high elevation precipitation volumes.**

**Given that temperatures remain close to 0°C during winter precipitation events, over large areas of Mount Lebanon, a better parameterization of the precipitation phase partitioning is expected to enhance the simulations and help in**

better capturing the rain on snow events. In general, this is probably a key issue for Mediterranean mountain regions with mild winters.”

[lines 491 – 498] **“The surface radiative forcing effect due to the deposition of mineral dust, such as those from the African Sahara and the Arabian Desert, is not included in SnowModel, but recent studies in regions with similar climate suggest that it can have a significant impact on snowmelt runoff (e.g. Painter et al., 2018). In fact, most of the energy used for snowmelt is shortwave radiation, with the occurrence of numerous dust storm events in this region, a proper parameterization of surface albedo, through assimilation of remote sensing products, as for example, could**

**help in better capturing the onset of snowmelt especially at mid elevations. Collecting information on the radiative and thermal fluxes and measuring the properties of the different snowpack layers could help in better representing these processes in models.”**

[R1] I also would like to know more about the improvement (or limitations) of the snow blowing and redistribution module used in the model. Does it really help to improve the spatial distribution of snow over the area?.

The discussion on the importance of blowing snow now reads as (lines 461 – 469): “**Blowing snow and its redistribution are important in this study domain. This is justified from the high variability of snow depth at shorter distances as concluded from the field observations conducted by the authors during two winter seasons. The existence of a large number of sinkholes in the region act as a trap for blowing snow. Figure 11 is an example where snow depth varies from zero to more than 2.5 m over distances as short as a few hundred meters due to snow transport by wind.**

**Previous work showed that the blowing snow and snow redistribution module in SnowModel improves the accuracy of the simulations (see Gascoin et al., 2013). Hence, we chose not to focus on this aspect in this study and assume that it contributes to a realistic representation of the snowpack. Capturing snow redistribution of snow in the future would probably require focusing on a smaller model domain with an accurate fine scale elevation DEM that is currently not available in this region. A proper representation of the snow redistribution could also help in better**

**explaining the role of advective heat fluxes and improve the model performance in capturing SWE.”**

[R1] Finally, **I also guess that sublimation is another important component of the SEB in Lebanon (as in other Mediterranean Mountains), what does the model inform about this process, is it an important source of uncertainty for snow modeling in this area?.**

We highlighted the importance of snow sublimates by adding (lines 471 – 484) and Figure 13: “**Snow sublimation can be a significant component of the snow mass balance as in other Mediterranean Mountains (e.g. Schulz and de Jong, 2004). In SnowModel there are two sources of sublimation: static surface sublimation and blowing snow sublimation. Figure 13 shows the distribution of the mean annual sublimated which reaches 70 mm we in the upper area of the study domain, and accounts for a fraction of about 5 to 15% of the annual peak SWE. Our justifications for the**

**relatively low sublimation rates over the study domain can be associated with the (1) high relative humidity due to the**

proximity to the Mediterranean Sea, which reduces both sources of sublimation (2) the high snow densification rates that inhibit blowing snow sublimation (Liston and Elder, 2006). We are unable to qualify the simulated sublimation however, the point (2) is compatible with our field observations that the snow surface at the mid-elevation is subject to multiple melt and refreeze events. Such snowpack surface limits the capacity for snow removal by wind and hence reduces snow sublimation. To conclude the model suggests that the sublimation is probably not the main source of uncertainty in this area (precipitation remains the main uncertainty), but we suggest this aspect should be the focus of further investigation using in situ measurements by eddy covariance towers or lysimeters."

[Figure]

Figure 13. Mean annual snow sublimation in [mm] (Pflug et al. 2019 model).

**Updated references**

Baba, M. W., Gascoin, S., Kinnard, C., Marchane, A., and Hanich, L.: Assimilation of Sentinel-2 Data into a Snowpack Model in the High Atlas of Morocco., Remote Sens, 10, 1982; doi:10.3390/rs10121982, 2018

Gascoin, S., Hagolle, O., Huc, M., Jarlan, L., Dejoux, J.-F., Szczypta, C., Marti, R., Sánchez, R.: A snow cover climatology for the Pyrenees from MODIS snow products, Hydrol. Earth Syst. Sci. 19, 2337–2351, 2015

Gascoin, S., Lhermitte, S., Kinnard, C., Bortels, K., and Liston, G. E.: Wind effects on snow cover in Pascua-Lama, Dry Andes of Chile. Advances in Water Resources, 55, 25-39, 2013.

Gorelick, N., Hancher, M., Dixon, M., Ilyushchenko, S., Thau, D., and Moore, R.: Google Earth Engine: Planetary-scale geospatial analysis for everyone. Remote Sensing of Environment, 202, 18-27, 2017.

Harder, P., and Pomeroy, J. W. (2014). Hydrological model uncertainty due to precipitation-phase partitioning methods. Hydrological Processes, 28(14), 4311-4327, 2014.

López-Moreno, J.I., Fassnacht, S.R., Beguería, S., Latron, J.: Variability of snow depth at the plot scale: implications
for mean depth estimation and sampling strategies. Cryosphere 5, 617–629, 2011.

López-Moreno, J.I., Fassnacht, S.R., Heath, J.T., Musselman, K.N., Revuelto, J., Latron, J., Morán-Tejeda, E., Jonas, T.: Small scale spatial variability of snow density and depth over complex alpine terrain: Implications for estimating snow water equivalent. Adv. Water Resour. 55, 40–52, 2013.

Nagler, T., Rott, H., Ripper, E., Bippus, G., and Hetzenecker, M.: Advancements for snowmelt monitoring by means
of sentinel-1 SAR. Remote Sensing, 8(4), 348, 2016.

Schulz, O., and de Jong, C.: Snowmelt and sublimation: field experiments and modelling in the High Atlas Mountains of Morocco, Hydrol. Earth Syst. Sci. 8,1076–1089, 2004.

Sturm, M., Taras, B., Liston, G., Derksen, C., Jonas, T., Lea, J.: Estimating snow water equivalent using snow depth data and climate classes. J. Hydrometeorol. 11, 1380–1394, 2010.

**Other edits**

We moved the figures showing the comparison of shortwave and albedo from the model evaluation section to a supplementary file as we found that such evaluation are no longer consistent with the emphasis made on water percolation (section 4.1). We updated section 4.1 (lines 310 – 314) to read: **" We note that we have also evaluated the modelled
shortwave radiation and snow albedo (see Figure. S1 in the Supplement). The results show that the monthly 
[revised manuscript text omitted]
., 2017). We did not use a more complex precipitation-phase partitioning method such as those described in Harder and Pomeroy (2014) and Harpold et al. (2017) because the parameterization of such methods would require introducing more parameters that we will not be able to determine for this study area. Second, we did not activate the default correction of the precipitation rate with elevation. It means that we ran the model with the minimal hypothesis that the precipitation spatial distribution is only controlled by the
distances with the AWS according to the Barnes interpolation scheme (Liston et al., 2006b). This approximation is justified by the good horizontal (Fig.Figure 1) and vertical distribution (Fig.Figure 2) of the AWS in the simulation domain and the fact that we could not identify a robust elevation effect from the AWS precipitation records.

The liquid water percolation scheme by Pflug et al. (2019) introduces an additional parameter, a freezing curve parameter after Jordan, (1991), that we left to the default value of 50 kg m$^{-3}$ following Pflug et al. (2019). The gravity-drainage scheme
implies to run the simulation with the multi-layer snowpack option that is not activated in the default run.

[Figure]

**Figure 2.** Elevation histogram of the simulation domain (in meters above mean sea level) including the position of the AWS. The dashed line indicates the reference elevation which is generally considered as the lower winter snowline elevation in Mount Lebanon, i.e. the
elevation below which the snow cover is virtually absent (1200 m).

**3.3 Model evaluation**

The validation dataset includes (1) half-hourly snow height (HS), snow albedo and incoming shortwave radiation collected at the AWS from W1314 to W1516, (2) bi-weekly manual SWE and snow density measurements collected near the AWS during the snow seasons of W1415 and W1516, and (3) daily snow cover area (SCA) observations from MODIS from W1314 to
W1516. All these data are fully described in Fayad et al. (2017b) and are available as open data in a public repository (Fayad et al., 2017c).

     Nearly continuous snow depth is available for water years W1415 and W1516 at LAQ and for water years W1314, W1415 and W1516 at MZA and CED. Half-hourly snow depth records were averaged to the daily timestep. We computed the daily albedo of the surface below AWS from half-hourly upward and downward-looking pyranometers measurements following
Stroeve et al. (2013), i.e. by computing the ratio of the daily incoming and reflected shortwave radiation totals. Summer albedo values ranged between 0.2 and 0.4 depending on the site and the year, hence we removed albedo values lower than 0.5 to keep only snow albedo measurements. The calculated daily snow albedo values and the incoming shortwave radiation were averaged to monthly values. The MODIS dataset is a daily cloud-free time series of snow cover maps providing the snow presence and absence at 500 m resolution (binary SCA). The model outputs at 100 m resolution were converted to binary SCA using a
threshold of 10 mm w.e. of SWE.

We used the snow course locations to approximate the pixels that are representative for each snow course. The simulated SWE were evaluated by comparing the observed SWE (revisit time ~ 11 days on average for all snow courses according to Fayad et al. (2017b)) with the simulated SWE value corresponding to the same dates when the in situ observations were taken. Five snow courses located near the three stations are used to showcase the evolution of SWE across different elevation bands (snow courses near the three AWS have elevations at 2823 and 2834 m a.s.l. for CED, 2297 and 2301 m a.s.l. for MZA, and 1843 m a.s.l. for LAQ).

To further investigate the effect of the liquid water percolation scheme on the internal properties of the snowpack, we used the wet snow detection from Sentinel-1 C-band Synthetic Aperture Radar data. We use these data as a proxy of melting conditions that we compared to the simulated release of liquid water from the snowpack. The wet snow detection was done using Sentinel-1 observations over the period October 2014 to June 2016. We extracted Sentinel-1 backscatter in VV polarization mode from the Sentinel-1 Ground Range Detected (GRD) collection in Google Earth Engine (Gorelick et al., 2017). We spatially averaged the backscatter at each station using a buffer with a radius of 100 m. Then, we defined a reference "dry" surface backscatter for each station using the 10th percentile of the backscatter time series (CED-8.4 dB, LAQ: -10.7

dB, MZA: -6.10 dB). A negative departure of 4 dB to this reference was used to determine the occurrences of wet snow (Nagler et al., 2016).

The main reason for using daily HS, SCA, and Sentinel-1 wet snow observations in the validation of the SWE is the scarcity of the spatially and temporally distributed SWE observations. However, HS is known to be a good proxy of SWE in many regions (Sturm et al., 2010). Furthermore, Fayad et al. (2017b) demonstrated that the estimation of SWE from HS could be achieved with acceptable accuracy using two years of observations in Mount Lebanon. Both the SWE and HS measurements are collected at point locations (Fayad et al., 2017c) and may not be representative at the catchment scale due to the high spatial variability of the snowpack in Mediterranean Mountains (e.g. López-Moreno et al., 2011, 2013). The use of SCA is justified in the absence of distributed SWE or HS measurements to validate the model (e.g., Gascoin et al., 2015; Baba et al., 2018).

**3.4 SWE estimation**

We used the model outputs to estimate the evolution of the SWE over the three basins (Fig. 1). The basins surface boundaries were used to estimate the SWE contribution for each basin by computing the mean daily SWE for all pixels within each basin. To account for the hydrological contribution of SWE from the snow dominated regions, the daily distributed SWE was spatially integrated for elevations above 1200 m a.s.l.. From the temporal evolution of these basin-scale SWE time series we derived the following key indicators: date and value of the peak SWE (maximum SWE during a water year), snow melt-out date (the first day of the calendar year on which SWE gets below 1 mm w.e.), distributed SWE for the peak SWE data (this map is important from a hydrological perspective as it shows the amount of snow available for melt at the end of the snowfall season for each hydrological year).

~~We used the model outputs to estimate the evolution of SWE over the three basins. The daily distributed SWE was spatially integrated over the area of each basin located above 1200 m a.s.l.. From the temporal evolution of these basin-scale SWE time series we derived the following key indicators: date and value of the peak SWE (maximum SWE during a water year), snow melt-out date (first day of the calendar year on which SWE gets below 1 mm w.e.).~~

**4 Results**

**4.1 Model evaluation**

Figure 3 compares the observed and modelled SWE evolution using both SnowModel configurations. The Pflug et al. (2019) model provides a better simulation of SWE during the melt season in CED and MZA. At LAQ, both models are biased to draw a robust conclusion on the effect of the percolation scheme. Both models are negatively biased in W1415 and either positively biased (default model) or negatively biased (Pflug et al., 2019) in W1516.

[Figure]

**Figure 3.** Time series of observed and modelled daily SWE at each AWS. SWE data were obtained from manual sampling during snow course measurements.

Figure 4 compares the modelled snow depth time series with the continuous snow depth measurements at each AWS. Although the difference between both models are less marked, similar observations as above can be made. This comparison also shows that the model reproduces well the snow depth evolution during W1314 and W1415 at CED and MZA, and the Pflug et al. (2019) model seems to be more consistent at these stations. The model overestimates snow depth at MZA during W1516 but the snow depth measurements are very noisy during this period suggesting that the data may be affected by a sensor issue, especially given the SWE observations presented above. Otherwise, the same positive bias as noted above (Fig. 3) can be observed at LAQ for W1415.

[Figure]

**Figure 4.** Time series of observed and modelled daily snow depth at each AWS.

The comparison of the modelled and observed snow cover shows that both models perform well in reproducing the snow cover evolution at the catchment scale, although the default model tend to overestimate the snow cover area during the ablation periods. This is particularly evident in spring 2016, but also during the atypical melt event in January 2014.

[Figure]

**Figure 5.** Time series of observed and modelled daily snow cover area in each study catchment.

We also evaluate the model using the snow cover duration from MODIS over the entire study area (Fig. 6). The results show that the spatial patterns of snow cover duration are generally better reproduced with the Pflug et al. (2019) scheme. The largest bias is found in the northernmost region, which corresponds to the region with the highest elevation (2700 to 3000 m a.s.l.). However, the average bias for this model configuration over the entire domain remains low (2 days).

~~Overall, Figures 3, 4 and 5 suggest that the Pflug et al., (2019) model provides a better representation of the snow cover in terms of snow depth, SWE and SCA. Table 3 shows that the correlation and RMSE are generally improved in comparison with the default model. The notable exceptions are small increases in the RMSE of snow depth and SWE at LAQ, which are hardly conclusive given the large accumulation biases at this AWS (see above, Figure 3 and Figure 4). At CED and MZA, the~~

**Table 3.** Performance of both model runs, default SnowModel and Pflug et al. (2019). Where, SD is daily snow depth from continuous acoustic gauges, SWE is snow water equivalent from snow course measurements, and SCA is daily snow cover area from MODIS. The value in bold indicate the best performance.

| | | Correlation (r) | | RMSE | |
| --- | --- | --- | --- | --- | --- |
| | | Default | Pflug | Default | Pflug |
| SD, [m] | CED | 0.73 | **0.84** | 0.67 | **0.38** |
| | LAQ | 0.72 | **0.76** | **0.32** | 0.38 |
| | MZA | 0.41 | **0.58** | 0.79 | **0.45** |
| SWE, [m] | CED | 0.28 | **0.78** | 0.30 | **0.22** |
| | LAQ | 0.39 | **0.56** | **0.21** | 0.26 |
| | MZA | 0.58 | **0.93** | 0.25 | **0.16** |
| SCA, [km⁻²] | Abou Ali | 0.89 | **0.91** | 56 | **40** |

| | | | | |
|---|---|---|---|---|
| Ibrahim | 0.90 | **0.93** | 49 | **36** |
| El Kelb | **0.89** | 0.88 | 32 | **30** |
| **Best model count** | **1** | **8** | **2** | **7** |

Finally, wWe also evaluate the model using the snow cover duration from MODIS over the entire study area (Figure 7). The results show that the spatial patterns of snow cover duration are generally well better reproduced with the Pflug et al. (2019) scheme. There isThe largest a positive bias is found in the northernmost region, which corresponds to the region with the highest elevation (2700 to 3000 m a.s.l.). However, the average bias for this model configuration over the entire domain remains low (2 days).

[Figure]

**Figure 6.** Scatterplots of observed vs. modelled monthly incoming shortwave radiation and snow albedo. Here the Pflug et al. (2019) model was used.

[Figure]

**Figure 6.** Top: difference in days between the simulated and MODIS SCD, computed over the entire period of the simulation (three snow seasons, from 01 November to 30 June, between 2013 and 2016) using the default SnowModel and the Pflug et al. (2019) model configurations. Bottom: the scatterplots show the pixel-by-pixel comparison of the simulated and MODIS SCD.

Interestingly, Figure 7 shows that the Pflug et al. (2019) scheme had a greater impact on the snow cover duration in the mid-elevation zones near 2000 m a.s.l. This is consistent with the expected behaviour of this scheme since these areas are prone to continuous melting during the winter and the Pflug et al. (2019) scheme is expected to have a greater impact in areas where the snowpack is wet and isothermal.

[Figure]

**Figure 7.** Left: map of the difference in days between the snow cover duration (SCD) simulated by the default model and the Pflug et al. (2019) model. Right: mean difference by elevation band. The SCD was computed over the simulation period (three snow seasons (from 01 November 2013 to 01 July 2016).

Figure 8 shows the time series of wet snow detection from Sentinel-1 over the period October 2014 to June 2016 and the simulated melt runoff for both model configurations. It can be observed that at CED and MZA, snowmelt runoff using the default model occurs over a short period of time at the end of the snow melt season, while the runoff simulated using the Pflug et al. (2019) model occurs earlier in the season and is more temporally distributed. This temporal distribution is consistent with Sentinel-1 data that indicate the presence of liquid water in the snowpack over the same periods. Again, this is consistent with the expected behaviour of the new liquid percolation scheme, which allows a more gradual release of liquid water throughout the melt season when the snowpack contains meltwater.

[Figure]

**Figure 8**: Time series of wet snow occurrences as detected from Sentinel-1 observations and modelled daily snowmelt runoff at each AWS.

Overall, all above results suggest that the Pflug et al., (2019) model provides a better representation of the snow cover in terms of snow depth, SWE and SCA. Table 3 shows that the correlation and RMSE are generally improved in comparison with the default model. The notable exceptions are small increases in the RMSE of snow depth and SWE at LAQ, which are hardly conclusive given the large accumulation biases at this AWS (Fig.ure 3 and FigureFig. 4). At CED and MZA, the performances of the model are significantly improved with the Pflug et al. (2019) version in terms of both correlation and RMSE with observed SWE.

We note that we have also evaluated the modelled shortwave radiation and snow albedo (see Figure. S1 in the Supplement). The results show that the monthly 
[revised manuscript text omitted]

- The comparison with MODIS data suggests that the enhanced model performed better at mid-elevations but there remains a positive bias in the snow cover duration at high elevation (Fig. 6). In this study, the model was forced with observations collected from three AWS but precipitation data were not available at the highest AWS (CED). Since then a precipitation gauge was installed at CED and high elevation precipitation data are available. Therefore, we suggest that the simulations could be improved by using observed precipitation from all stations to have a better representation of the high elevation precipitation volumes.

- Given that temperatures remain close to 0°C during winter precipitation events, over large areas of Mount Lebanon, a better parameterization of the precipitation phase partitioning is expected to enhance the simulations and help in better capturing the rain on snow events. In general, this is probably a key issue for Mediterranean mountain regions with mild winters.

- Blowing snow and its redistribution are important in this study domain. This is justified from the high variability of snow depth at shorter distances as concluded from the field observations conducted by the authors during two winter seasons. The existence of a large number of sinkholes in the region act as a trap for blowing snow. Figure 11 is an example where snow depth varies from zero to more than 2.5 m over distances as short as a few hundred meters due to snow transport by wind. Previous work showed that the blowing snow and snow redistribution module in SnowModel improves the accuracy of the simulations (see Gascoin et al., 2013). Hence, we chose not to focus on this aspect in this study and assume that it contributes to a realistic representation of the snowpack. Capturing snow redistribution of snow in the future would probably require focusing on a smaller model domain with an accurate fine scale elevation DEM that is currently not available in this region. A proper representation of the snow redistribution could also help in better explaining the role of advective heat fluxes and improve the model performance in capturing SWE.

- Snow sublimation can be a significant component of the snow mass balance as in other Mediterranean Mountains (e.g. Schulz and de Jong, 2004). In SnowModel there are two sources of sublimation: static surface sublimation and blowing snow sublimation. Figure 13 shows the distribution of the mean annual sublimated which reaches 70 mm we in the upper area of the study domain, and accounts for a fraction of about 5 to 15% of the annual peak SWE. Our justifications for the relatively low sublimation rates over the study domain can be associated with the (1) high relative humidity due to the proximity to the Mediterranean Sea, which reduces both sources of sublimation (2) the high snow densification rates that inhibit blowing snow sublimation (Liston and Elder, 2006). We are unable to qualify the simulated sublimation however, the point (2) is compatible with our field observations that the snow surface at the mid-elevation is subject to multiple melt and refreeze events. Such snowpack surface limits the capacity for snow removal by wind and hence reduces snow sublimation. To conclude the model suggests that the sublimation is probably not the main source of uncertainty in this area (precipitation remains the main uncertainty), but we suggest this aspect should be the focus of further investigation using in situ measurements by eddy covariance towers or lysimeters.

[Figure]

**Figure 13.** Mean annual snow sublimation in [mm] (Pflug et al. 2019 model).

-      The surface radiative forcing effect due to the deposition of mineral dust, such as those from the African Sahara and the Arabian Desert, is not included in SnowModel, but recent studies in regions with similar climate suggest that it can have a significant impact on snowmelt runoff (e.g. Painter et al., 2018). In fact, most of the energy used for snowmelt is shortwave radiation, with the occurrence of numerous dust storm events in this region, a proper parameterization of surface albedo, through assimilation of remote sensing products, as for example, could help in better capturing the onset of snowmelt especially at mid elevations. Collecting information on the radiative and thermal fluxes and measuring the properties of the different snowpack layers could help in better representing these processes in models.

[revised manuscript text omitted]

Lebanese Ministry of Energy and Water: Lebanese National Water Sector Strategy (as approved by the Lebanese Government resolution No 2. Date: March 09 2012). Available at: http://www.databank.com.lb/docs/National%20Water%20Sector%20Strategy%202010-2020.pdf (last access: 12 July 2019),
2012.

Lebanese Ministry of Environment (MOE/UNDP/ECODIT): State and Trends of the Lebanese Environment. Available at http://www.undp.org.lb/communication/publications/downloads/SOER_en.pdf (last access: 12 July 2019), 2011.

Liston, G. E., and Elder, K.: A distributed snow-evolution modeling system (SnowModel), Journal of Hydrometeorology, 7(6), 1259-1276, 2006a.

Liston, G. E., and Elder, K.: A meteorological distribution system for high-resolution terrestrial modeling (MicroMet), Journal of Hydrometeorology, 7(2), 217-234, 2006b.

Liston, G. E., and Hiemstra, C. A.: A simple data assimilation system for complex snow distributions (SnowAssim), Journal of Hydrometeorology, 9(5), 989-1004, 2008.

Liston, G. E., and Mernild, S. H.: Greenland freshwater runoff. Part I: A runoff routing model for glaciated and nonglaciated
landscapes (HydroFlow), Journal of Climate, 25(17), 5997-6014, 2012.

Liston, G. E., Haehnel, R. B., Sturm, M., Hiemstra, C. A., Berezovskaya, S., and Tabler, R. D.: Simulating complex snow distributions in windy environments using SnowTran-3D, Journal of Glaciology, 53(181), 241-256, 2007.

Liston, G. E., Pielke, R. A., and Greene, E. M.: Improving first order snow related deficiencies in a regional climate model, Journal of Geophysical Research: Atmospheres, 104(D16), 19559-19567, 1999.

Liston, G. E.: Local advection of momentum, heat, and moisture during the melt of patchy snow covers, J. Appl. Meteor., 34, 1705–1715, 1995.

Loarie, S., Duffy, P., Hamilton, H., Asner, G., Field, C., and Ackerly, D.: The velocity of climate change, Nature, 462, 1052–1055, 2009.

López-Moreno, J.I., Fassnacht, S.R., Beguería, S., Latron, J.: Variability of snow depth at the plot scale: implications for mean
depth estimation and sampling strategies. Cryosphere 5, 617–629, 2011.

López-Moreno, J.I., Fassnacht, S.R., Heath, J.T., Musselman, K.N., Revuelto, J., Latron, J., Morán-Tejeda, E., Jonas, T.: Small scale spatial variability of snow density and depth over complex alpine terrain: Implications for estimating snow water equivalent. Adv. Water Resour. 55, 40–52, 2013.

Margane, A., Schuler, P., Königer, P., Abi Rizk, J., Stoeckl, L., and Raad, R.: Hydrogeology of the Groundwater Contribution
Zone of Jeita Spring, Technical Cooperation Project Protection of Jeita Spring, BGR Technical Report No. 5, 317 pp., Raifoun, Lebanon, 2013.

Marti, R., Gascoin, S., Berthier, E., Pinel, M. D., Houet, T., and Laffly, D.: Mapping snow depth in open alpine terrain from stereo satellite imagery, The Cryosphere, 10(4), 1361-1380, 2016.

Mernild, S. H., Liston, G. E., Hiemstra, C. A., Malmros, J. K., Yde, J. C., and McPhee, J.: The Andes Cordillera. Part I: snow distribution, properties, and trends (1979–2014), International Journal of Climatology, 37(4), 1680-1698, 2017.

Mernild, S.H., Hasholt, B., and Liston, G.E.: Climatic control on river discharge simulations, Zackenberg River drainage basin, northeast Greenland, Hydrol. Process., 22, 1932–1948, 2008.

Mhawej, M., Faour, G., Fayad, A., and Shaban, A.: Towards an enhanced method to map snow cover areas and derive snow-water equivalent in Lebanon, J. Hydrol., 513, 274–282, https://doi.org/10.1016/j.jhydrol.2014.03.058, 2014.

Mott, R., Vionnet, V., and Grünewald, T.: The seasonal snow cover dynamics: review on wind-driven coupling processes, Frontiers in Earth Science, 6, 197, 2018.

Musselman, K. N., Molotch, N. P., and Margulis, S. A.: Snowmelt response to simulated warming across a large elevation gradient, southern Sierra Nevada, California, The Cryosphere, 11(6), 2017.

Nagler, T., Rott, H., Ripper, E., Bippus, G., and Hetzenecker, M.: Advancements for snowmelt monitoring by means of sentinel-1 SAR. Remote Sensing, 8(4), 348, 2016.

National Council for Scientific Research (NCRS): Land-use land cover map of Lebanon, Beirut, Lebanon, available at: http://www.cnrs.edu.lb/ (last access: 12 July 2019), 2015.

Nohara, D., Kitoh, A., Hosaka, M., and Oki, T.: Impact of climate change on river discharge projected by multimodel ensemble, J. Hydrometeorol., 7, 1076–1089, 2006.

Painter, T. H., Skiles, S. M., Deems, J. S., Brandt, W. T., and Dozier, J.: Variation in rising limb of Colorado River snowmelt runoff hydrograph controlled by dust radiative forcing in snow, Geophysical Research Letters, 45, 797– 808. https://doi-org.insu.bib.cnrs.fr/10.1002/2017GL075826, 2018.

Pflug, J. M., Liston, G. E., Nijssen, B., and Lundquist, J. D.: Testing model representations of snowpack liquid water percolation across multiple climates, Water Resources Research, 55(6), 4820-4838, 2019.

Schulz, O., and de Jong, C.: Snowmelt and sublimation: field experiments and modelling in the High Atlas Mountains of Morocco, Hydrol. Earth Syst. Sci. 8,1076–1089, 2004.

Shaban, A., Faour, G., Khawlie, M., and Abdallah, C.: Remote sensing application to estimate the volume of water in the form of snow on Mount Lebanon/Application de la télédétection à l'estimation du volume d'eau sous forme de neige sur le Mont Liban, Hydrolog. Sci. J., 49, 643–653, https://doi.org/10.1623/hysj.49.4.643.54432, 2004.

Sproles, E.A., Nolin, A.W., Rittger, K., Painter, T.: Climate change impacts on maritime mountain snowpack in the Oregon Cascades, Hydrol. Earth Syst. Sci., 17, 2581–2597, 2013.

Stroeve, J., Box, J.E., Wang, Z., Schaaf, C., and Barrett, A.: Re-evaluation of MODIS MCD43 Greenland albedo accuracy and trends, Remote Sens. Environ, 138, 199–214, 2013.

Sturm, M., Holmgren, J., and Liston, G. E.: A seasonal snow cover classification system for local to global applications,

Journal of Climate, 8(5), 1261-1283, 1995.

Sturm, M., Taras, B., Liston, G., Derksen, C., Jonas, T., Lea, J.: Estimating snow water equivalent using snow depth data and climate classes. J. Hydrometeorol. 11, 1380–1394, 2010.

Telesca, L., Shaban, A., Gascoin, S., Darwich, T., Drapeau, L., Hage, M., and Faour, G.: Characterization of the time dynamics of monthly satellite snow cover data on Mountain Chains in Lebanon, J. Hydrol., 519, 3214–3222, https://doi.org/10.1016/j.jhydrol.2014.10.037, 2014.

UN Development Programme (UNDP): Assessment of Groundwater Resources of Lebanon, Beirut, available at: http://www.lb.undp.org/content/lebanon/en/home/library/environment_energy/assessment-of-groundwater-resources-of-lebanon.html (lastaccess: 7 August 2017), 2014.

Viviroli, D., Dürr, H. H., Messerli, B., Meybeck, M., and Weingartner, R.: Mountains of the world, water towers for humanity: Typology, mapping, and global significance, Water Resources Research, 43(7), 2007.